



# Seasonal streamflow forecasting by conditioning climatology with precipitation indices

Louise Crochemore[1], Maria-Helena Ramos[1], Florian Pappenberger[2], Charles Perrin[1]

[1]IRSTEA, Catchment Hydrology Research Group, UR HBAN, Antony, France
[2]ECMWF, European Centre for Medium-Range Weather Forecasts, Shinfield Park, Reading, UK

*Correspondence to*: Louise Crochemore (louise.crochemore@irstea.fr)

**Abstract.** Many fields such as drought risk assessment or reservoir management can benefit from long-range streamflow forecasts. The simplest way to make probabilistic streamflow forecasts can be to use historical streamflow time series, if available. Another approach is to use ensemble climate scenarios as input to a hydrological model. Climatology (i.e. time
series of climate conditions recorded over a long time period) has long been used in long-range streamflow forecasting. However, in the last decade, the use of general circulation model (GCM) outputs as input to hydrological models has developed. While precipitation climatology and historical streamflows offer reliable ensembles, forecasts based on GCM outputs can offer sharper ensembles, partly due to the initialisation of GCMs and hydrological models on current conditions. This study proposes to condition historical data based on GCM precipitation forecasts to get the most out of both data
sources and improve seasonal streamflow forecasting in France. Four conditioning statistics based on ECMWF System 4 forecasts of cumulative precipitation and of the Standardized Precipitation Index (SPI) were used to select traces within historical streamflows and historical precipitations. The four conditioned precipitation scenarios were used as input to the GR6J hydrological model to obtain eight conditioned streamflow forecast scenarios. These streamflow scenarios were compared to three references: an ensemble based on historical streamflows, the widespread Ensemble Streamflow Prediction
(ESP) ensemble, and System 4 precipitation forecasts. These ensembles were evaluated based on their sharpness, reliability and overall performance.

An overall comparison of forecast ensembles showed that conditioning past observations based on the three-month Standardized Precipitation Index (SPI3) improved the sharpness of ensembles based on historical data, while maintaining a good reliability. An evaluation of forecast ensembles for low-flow forecasting showed that the SPI3-conditioned ensembles
provided reliable forecasts of low-flow duration and deficit volume based on the 80[th] exceedance percentile. Last, drought risk forecasting was illustrated for the 2003 drought.





## 1 Introduction

### 1.1 Approaches to seasonal forecasting

Numerical prediction is valuable to proactively manage risks in areas such as hydropower, drinking water production and drought preparedness (Wilhite et al., 2000). Regardless of the application, probabilistic forecasts are preferred over

deterministic ones to convey uncertainties (Krzysztofowicz, 2001; Ramos et al., 2013). The main sources of uncertainty in informing decision-making depend on the variable being forecast, the forecast horizon, but also on the location. For instance, region-specific tools have been developed in the world to predict and anticipate drought events weeks, months or even years in advance (Anderson et al., 2000; Ceppi et al., 2014; Hao et al., 2014; Sheffield et al., 2013; Shukla et al., 2014). Nevertheless, anticipating river runoff events at long lead times remains a challenge (Yuan et al., 2015).

The predictability of streamflow at long lead times lies in the initial hydrological conditions and the meteorological forcing. Research has shown that the relative role of each source of predictability mainly depends on the "inertia" or "memory" of the studied basin, the forecast season and the forecast lead time (Shukla et al., 2013; Wood and Lettenmaier, 2008; Yossef et al., 2013). Yossef et al. (2013) showed that in Western Europe, from July to October, streamflow forecasts are more dependent on meteorological forcing than they are on initial conditions, even one month ahead. The conclusions of Shukla et al. (2013)

are quite consistent with these findings. They found that the predictability of a forecast issued in July in France lies in the meteorological forcing for horizons longer than three months. However, at shorter lead times, their results show that predictability can be led by either initial conditions or meteorological forcing, depending on the geographical location in France.

In practice, two approaches are often used to forecast streamflow at the seasonal scale (Easey et al., 2006). Statistical

approaches rely on past observations and statistical relationships between a predictor and a predictand. Dynamical approaches rely on coupled general circulation model (GCM) outputs or past observations to feed a hydrological rainfall-runoff model. The choice of one approach over the other will depend on the purpose of the forecast, the region of interest and on the available data. More importantly, some studies have shown that the two approaches can complement and benefit from each other (Block and Rajagopalan, 2009; Seibert and Trambauer, 2015).

Climatology (past observations) is considered a good indicator of the range of possible outcomes for a given time of the year. Day (1985) introduced the Ensemble Streamflow Prediction (ESP), which is an approach that uses precipitation climatology as input to a hydrological model previously initialised for the forecast date. This approach has been extensively used, for research purposes and operationally, in seasonal streamflow forecasting (Wang et al., 2011) and reservoir operations (Faber and Stedinger, 2001), among other fields. An alternative to climatology is the seasonal forecasts issued by

GCMs (Yuan et al., 2015). While these are initialized and forced for a specific forecast day, precipitation climatology simply provides a range of what has been previously observed on the forecast day, regardless of the current atmospheric situation and latest observations.





## 1.2 Selecting ensembles to improve long-range forecasting

More recently, research has focused on fine-tuning the traditional ESP method by selecting relevant years within the climatology. In that context, several studies have proposed to condition or weight past observations based on climate signals. The proposed approaches are commonly divided in pre-ESP (prior to hydrological modelling, i.e. by conditioning climate ensembles) and post-ESP approaches (after hydrological modelling, i.e. by conditioning streamflow ensembles). In Northern America, several studies have taken advantage of the influence of the El Niño Southern Oscillation (ENSO) and the Pacific Decadal Oscillation (PDO) to improve the skill of seasonal forecasts. Hamlet and Lettenmaier (1999) selected past precipitations based on categories of ENSO and PDO to feed a hydrological model for streamflow forecasting, and, later on, for reservoir operation (Hamlet et al., 2002). Werner et al. (2004) selected and weighted traces based on the ENSO before and after hydrological modelling. The authors showed that the post-ESP method yielded greater improvements in forecast skill than the pre-ESP method. Their post-ESP method was recently applied by Trambauer et al. (2015) in Southern Africa. Gobena and Gan (2010) used the PDO in several pre- and post-ESP resampling, including a pre-ESP approach benefiting from monthly precipitation and temperature statistically derived from climate model outputs. Recent studies have investigated the use of multiple other climate indices in post-ESP techniques (Najafi et al., 2012). At the scale of the globe, van Dijk et al. (2013) selected traces within precipitation climatology based on climate indicators that were proven influential for the region and time period. They showed that using climate information improved forecast skill in Southeast Asia and South America.

In Europe, teleconnections show complex patterns and strongly depend on the season (Ionita et al., 2015). Bierkens and van Beek (2009) exploited the teleconnection found between winter precipitations and the Northern Atlantic Oscillation (NAO) to select traces within the precipitation climatology and forecast seasonal streamflows. In Czech Republic, Šípek et Daňhelka (2015) ran a hydrological model with synthetic series of precipitation and temperature generated from climate forecasts and historical meteorological series. In France, Sauquet et al. (2008) forecast low flows in the Rhine river by selecting past precipitation scenarios that were close to the forecast day in terms of previous amounts of precipitation. Other approaches have consisted in directly extracting information from long streamflow records. For instance, Svensson (2016) selected analogues within historical streamflows based on the streamflow anomaly observed in the month prior to the forecast date. The author aimed to forecast mean streamflow over the coming month or the coming three months in the United Kingdom.

In California, Carpenter and Georgakakos (2001) and Yao and Georgakakos (2001) tested several streamflow forecasting methods to forecast the inflows to the Folsom Lake. Based on the hypothesis that "*it is not necessary [...] that low skill in reproducing regional precipitation is an index of the utility of GCM information for systems acting as low-pass filters, such as the hydrological and reservoir systems are*", Carpenter and Georgakakos (2001) conditioned historical precipitations based on the precipitation anomaly forecast by a GCM. They found that this conditioning was particularly efficient to forecast the low 30-day inflows to the lake: "*Global climate model information from the Canadian coupled global climate model CGCM1 benefits the mean forecasts significantly mainly for low observed 30-day inflow volumes.*" Yao and



Georgakakos (2001) compared this method with the ESP method, and with a forecast ensemble conditioned from historical streamflows based on the latest observed reservoir inflows. They found that the GCM-conditioned ensemble outperformed the ESP method, although the ensemble conditioned from historical streamflows, which was the most reliable, managed to completely eliminate flood damage and generate more energy than the other two ensembles.

## 1.3 Scope of the study

This study proposes to investigate how selecting historical data based on forecast precipitation indices contributes to the skill of seasonal streamflow forecasts. Our approach selects traces of past observed precipitations and streamflows based on precipitation indices derived from the System 4 seasonal precipitation forecasts issued by the European Centre for Medium-range Weather Forecasts (ECMWF). The aim is to generate forecasts that benefit from the reliability of climatology-based ensembles and the sharpness of System 4 precipitation forecasts. In a previous study (Crochemore et al., 2016), we assessed the performance of System 4 precipitation forecasts for seasonal streamflow forecasting. Despite the good overall performance of the streamflow forecasts after bias correction, we still observed a lack of reliability of the forecasts generated with the hydrological model in summer. In accordance with the results from Carpenter and Georgakakos (2001), we evaluate the proposed methods in contexts of low flows and droughts.

Section 2 presents the data and the methodology used to build streamflow forecasts. In Section 3, we present the evaluation of the different studied scenarios. First, we analyse the impact of the conditioning on the overall performance, sharpness and reliability of seasonal streamflow forecasts over the whole year. Then, we investigate the discrimination and reliability of the ensemble prediction systems to forecast low-flow events. We also illustrate the performance of our approach in forecasting drought risks through the case of the 2003 severe drought in France. In Section 4, we discuss the main outcomes and perspectives of the study.

## 2 Data and methods

### 2.1 Observed and forecast hydrometeorological data

Observed precipitation data used in this study come from the SAFRAN reanalysis of Météo-France (Quintana-Seguí et al., 2008; Vidal et al., 2010). Daily values are available from August 1958 until July 2010 (i.e. 51 complete years) at an 8x8 km grid resolution covering France. Data were aggregated at the catchment scale. Mean areal potential evapotranspiration was computed for each catchment using a temperature-based formula (Oudin et al., 2005) and observed temperatures from the SAFRAN reanalysis. Daily streamflow data at the outlet of each catchment come from the French national archive (Banque Hydro, www.hydro.eaufrance.fr).

Seasonal precipitation forecasts used in this study were collected from ECMWF GCM, System 4. Once a month, ECMWF provides a 51-member forecast ensemble for the next seven months at a $T_L255$ (~0.7°) spatial resolution (Molteni et al.,





2011). ECMWF produced hindcasts from 1981 to 2010. These hindcasts are composed of 51 members when issued in February, May, August and November, and 15 members in other months. For the purpose of this study, System 4 forecasts were aggregated at the catchment scale. Only forecasts with lead times up to 90-days were considered. In a previous study, several bias corrections were applied to System 4 precipitation forecasts and compared based on their impact on seasonal

streamflow forecasting (Crochemore et al., 2016). The study showed that the empirical distribution mapping of daily values improved the reliability of both precipitation and streamflow forecasts. Following these results, System 4 precipitation forecasts used in this study were previously bias corrected with the empirical distribution mapping of daily values.

## 2.2 Catchments and hydrological model

Sixteen catchments spread over France were selected from the database used by Nicolle et al. (2014). Using a set of

catchments helps getting more general conclusions (see e.g. Andréassian et al., 2009; Gupta et al., 2014). However, it should be noted that identifying relations between performances and catchment characteristics is outside the scope of this study. These catchments are dominated by a pluvial regime and the quality of their streamflow data during low flows is good. The selected catchments additionally have an average solid fraction of precipitation below 10%. Their location, hydrological regimes and main characteristics are presented in Figure 1 and Table 1, respectively. In these catchments, low flows are

observed between May and October, i.e. from late spring to early autumn. Major drought events in these catchments include the droughts of 1976, 1989, 2003 and 2005. Among these, the 2003 drought was estimated to have caused 15,000 deaths and cost over a billion euros just in France (UNEP, 2004; Poumadère et al., 2005).

The hydrological model used in this study is the GR6J model, a daily conceptual model with six free parameters specifically proposed for low-flow simulation by Pushpalatha et al. (2011). The model is composed of three reservoirs (one for the

production function and two for the routing function), and one unit hydrograph to account for flow delays. Its inputs are daily precipitation and potential evapotranspiration at the catchment scale and its output is the streamflow at the catchment outlet. In this study, the mean interannual potential evapotranspiration was systematically used as input to the GR6J model. For a given day of the year, the estimated potential evapotranspiration on this day is assumed to be the mean of all potential evapotranspiration computed for this day of the year, from 1958 to 2010. Regardless of the precipitation scenario fed to the

model, the interannual potential evapotranspiration scenario is systematically used as input to the model so as to focus solely on the influence of precipitation inputs on streamflow forecasts. In addition, when the model is applied to forecast streamflow, the last observed streamflow at the time of forecast is used to update the levels of the routing reservoirs before issuing the forecasts.

The GR6J model was calibrated in each catchment with the one-year-leave-out method (Arlot and Celisse, 2010) and with

the Kling-Gupta Efficiency (Gupta et al., 2009) applied to inverse flows to focus on the lowest flows (Pushpalatha et al., 2012). We obtained an average KGE applied to inverse flows of 0.78 in calibration and 0.76 in validation over the sixteen catchments. An average KGE applied to flows of 0.78 was obtained in validation, showing that the model also performs well





for median to high flows. The distance of the bias from 1 (1-bias) is moderate in simulation with values ranging from -0.1 to 0.1 in all catchments but three. In these three catchments, values of 0.12, -0.14 and -0.94 are obtained.

## 2.3 Forecast verification methods

Many criteria exist to assess the performance of probabilistic forecasts. Here, we assessed their sharpness and reliability

following the paradigm introduced by Gneiting et al. (2007), that is maximizing sharpness while guaranteeing reliability. The overall performance and the discrimination of the forecasts were also evaluated.

### 2.3.1 Evaluation criteria

The overall performance of the forecast systems was evaluated by means of the Continuous Rank Probability Score (CRPS, Hersbach, 2000). The CRPS averages over the evaluation period the area between the cumulative forecast distribution and

the step function corresponding to the observation.

Sharpness is an intrinsic attribute of the forecast ensemble. It indicates how spread the members of an ensemble forecast are. Here, sharpness was computed as the average over the evaluation period of the difference between the 95[th] and the 5[th] percentiles of the forecast distribution (Gneiting et al., 2007). It thus corresponds to the 90% interquantile range (IQR).

Reliability refers to the statistical consistency between observed frequencies and forecast probabilities. Reliability was

evaluated with the Probability Integral Transform diagram (PIT, Gneiting et al., 2007; Laio and Tamea, 2007). The PIT diagram represents the cumulative distribution of the positions of the observation within the distribution of forecast values. The PIT diagram of a perfectly reliable forecast is superposed with the 1:1 diagonal, meaning that the observation uniformly falls within the forecast distribution. To numerically compare results for large datasets, Renard et al. (2010) proposed to compute the area between the PIT diagram and the 1:1 diagonal. The smaller the PIT area, the more reliable the ensemble.

The Relative Operating Characteristics diagram (ROC, Mason and Graham, 1999) is used to assess the capacity of forecasting systems to discriminate between events and non-events. In this study, the threshold used to define events is the 80[th] exceedance percentile of observed streamflow. To build the diagram, the proportion of ensemble members below the threshold necessary to trigger an alert varies from none to all ensemble members. For each of these proportions, the probability of detection is plotted against the false alarm ratio. The ROC diagram is plotted for a given threshold, catchment

and forecast lead time. The Area Under the Curve (AUC) summarizes the ROC diagram into one numerical value that allows for an easier comparison of forecast systems. The closer the AUC is to 1, the better the system is at discriminating between events and non-events.

### 2.3.2 Skill scores

The skill of forecast systems is computed as follows:





$$Skill\ score_i = \frac{Score_i^{ref} - Score_i^{syst}}{Score_i^{ref} + Score_i^{syst}} \qquad (1)$$

This normalized skill ranges within [-1,1]. A skill superior to 0 (inferior to 0) indicates that the forecast system performs better (worse) than the reference. The skill score was computed based on the CRPS, the IQR and the PIT area. These scores are abbreviated CRPSS, IQRSS and PITSS. Three base ensembles (see next section) were used in turn as reference forecasts,

to assess the skill of built forecast scenarios. Since we compared ensembles with different ensemble sizes (see Table 2), which is known to induce bias when computing skill scores, the correction proposed by Ferro et al. (2008) was applied to remove such bias in the computation of the CRPSS.

## 2.4 Forecast scenario building method

Eleven ensemble forecast scenarios were compared based on their performance in forecasting streamflows. Three scenarios

are based on methods commonly used in seasonal streamflow forecasting. These are named "base ensembles" in the following. The remaining eight scenarios are based on these base ensembles and specific conditioning statistics. Table 2 summarizes the different ensemble forecast scenarios compared in this study.

### 2.4.1 Description of base ensembles

The simplest ensemble forecast scenario uses the long-term statistical variability of historical streamflows. It is assumed that

the streamflow at a given day of the year is likely to fall within the range of streamflows observed in other years, on that same day. Apart from the necessity to have a long time series of streamflow records, this ensemble is not computationally costly. It is named **HistQ** hereafter.

Another base ensemble is the traditional **ESP** method. It requires a hydrological model and a long time series of precipitation records. This ensemble is based on the assumption that the precipitation of a given day is likely to fall within the range of

past precipitations observed in previous years, on that same day. For a given forecast day, a precipitation ensemble is thus built by using precipitations observed in other years. The precipitation ensemble has as many members as the number of years different from the forecast year available in the precipitation record. The states of the GR6J hydrological model are first initialized with a one year run up to the forecast date. The precipitation ensemble and interannual potential evapotranspiration are then used as input to the model.

The third base ensemble is similar to ESP but uses the bias corrected ECMWF System 4 seasonal precipitation forecasts as input to the GR6J hydrological model. Both the System 4 GCM and the hydrological model are initialized for the forecast day. This ensemble can be considered the most costly in terms of implementation and computational needs. Hereafter, this ensemble is named **Sys4**.



### 2.4.2 Description of conditioned scenarios

From the base ensembles, we built eight other scenarios by selecting traces within the HistQ and the ESP ensembles. The conditioning was based on statistics derived from the System 4 precipitation forecasts. Four statistics were computed for each forecast date and each member of the seasonal forecasting system. Two are based on cumulative rainfalls, and two on

the standardized precipitation index (SPI). The SPI transforms the distribution fitted to a long precipitation record into a normal distribution (McKee et al., 1993; WMO, 2012). An SPI value of 0 corresponds to conditions close to the long-term average of precipitations. Negative (positive) SPI values correspond to drier (wetter) conditions. The four conditioning statistics are:

- the cumulative precipitation forecast over the first three months of lead time (**Sum3**);

- the series of cumulative precipitation forecast over the first, second and third months (i.e. one value per lead time, **Sum1**);

- the SPI over the first three months altogether (**SPI3**);

- the SPI over the first, second and third months separately (i.e. one value per lead time, **SPI1**).

The statistics (SPI or precipitation volume) derived from System 4 forecasts are then used to select traces within HistQ and

ESP. For that purpose, statistics are also computed for sequences of historical precipitations. Here, we consider sequences that start within 15 days of the forecast date, observed in years different from the forecast year. For a given forecast member, the sequence that is the closest in terms of the Euclidian distance, and with regard to the considered statistics, is selected. Note that different forecast members can be associated with the same "closest" historical sequence.

Once the sequences are selected, two options can then lead to a streamflow forecast ensemble: (a) the selected precipitation

sequences can be used as input to the hydrological model to generate a streamflow forecast ensemble (**ESP_Sum3**, **ESP_Sum1**, **ESP_SPI3**, **ESP_SPI1**), or (b) the historical streamflows corresponding to the selected sequences can be directly used as ensemble members to build a streamflow ensemble (**HistQ_Sum3**, **HistQ_Sum1**, **HistQ_SPI3**, **HistQ_SPI1**). In the latter case, conditioning streamflow sequences based on rainfall statistics may result in unrealistic forecasts due to initial conditions far from what is observed on the forecast date. Therefore, when directly selecting scenarios

from past streamflow observations, the last observed streamflow is added as a conditioning criterion in the computation of the Euclidian distance.

## 3 Performance of the streamflow forecasting systems

### 3.1 Skill of System 4 in forecasting conditioning statistics

Before evaluating the performance of the eleven ensemble forecast scenarios, we evaluated the skill of System 4 in

forecasting the conditioning statistics (cumulative precipitations and SPI). Figure 2 shows their skill in overall performance (CRPSS) and in sharpness (IQRSS), and Figure 3 shows their reliability (PIT diagram). The reference forecast used to





compute the skill scores is historical precipitations (i.e. climatology). Regardless of the considered statistic, System 4 performs as well as climatology while being sharper. In addition, SPI forecasts issued from System 4 are reliable overall and in standard precipitation conditions. In dry conditions (i.e. SPI values smaller than -1), however, forecasts tend to overestimate SPI values, while in wet conditions (i.e. SPI values greater than 1) forecasts tend to underestimate SPI values.

5 Similar PIT diagrams are observed with SPI forecasts from historical precipitations (not shown). Dutra et al. (2014) did a similar comparison and showed that SPI forecasts from System 4 always had skill as compared to historical precipitations, with respect to discrimination, accuracy and anomaly correlation, in South Africa.

## 3.2 Statistical evaluation of accuracy and reliability

### 3.2.1 Influence of conditioning on streamflow forecasts performance

10 We evaluated the gain and loss in skill of daily streamflow forecasts due to the four types of conditioning applied to the HistQ base ensemble. Figure 4 shows the CRPSS, IQRSS and PITSS for lead times up to 90 days, and the PIT diagram for a lead time of 45 days. The reference for the computation of the skill is HistQ, i.e. historical streamflows with all available years. Each line corresponds to one of the 16 catchments.

The first conclusion from this figure is that all four conditionings lead to similar results. Their impact on forecasts reliability 15 (PIT) and sharpness (IQR) is uniform over the lead times, while their impact on overall performance (CRPS) is greater at shorter lead times. Conditioning HistQ improves sharpness at most lead times (IQRSS above zero) and, for all conditioning statistics (Sum or SPI). However, as a direct result of narrower ensembles, there is a decrease in the PIT values (reliability) at most lead times (PITSS below zero). Nevertheless, the PIT diagrams at 45 days show that this decrease does not affect the overall reliability of the conditioned ensembles: they remain quite reliable (PIT values close to the diagonal line) for all 20 conditioning statistics, especially when conditioning based on the SPI. Regarding overall performance (CRPS), the conditioning increases performance up to 15 to 30 days ahead in most catchments. Improvement is greater when traces are selected based on cumulative precipitations (Sum3 or Sum1) or SPI3 than when they are selected based on the series of SPI1 values. This improvement in overall performance in the first lead times can be attributed to the fact that the conditioning of historical streamflow takes into account the last observed streamflow. At longer lead times, the overall performance of 25 conditioned scenarios is, in the majority of catchments, equivalent or slightly worse than that of HistQ. In one of the catchments, however, we observed improvements up to 90 days ahead. This catchment corresponds to catchment 1, in which interannual streamflow variability dominates over seasonality (cf. Section 2.2) due to a high base flow index.

We also examined the loss and gain in skill due to conditioning the ESP base ensemble. Figure 5 is similar to Figure 4. It plots the skill scores against lead time and the PIT diagram for a lead time of 45 days. This time, the reference used in the 30 computation of the skill is ESP. Here again, the four conditionings seem to have a similar impact on performance. Conditioned streamflow forecasts appear to be as performant or slightly worse than ESP in terms of overall performance (CRPSS), for all lead times. This often translates in a gain in sharpness (IQRSS) associated with a loss in reliability (PITSS),





as observed with the scenarios conditioned from the HistQ base ensemble. Some distinctions between the conditionings based on cumulative precipitations and the conditionings based on the SPI can be seen. First, conditionings based on the SPI provide more homogeneous results between catchments for all evaluation criteria. We also observe that the loss in overall performance is greater with the conditionings based on cumulative precipitations, while overall performance of the

ensembles conditioned with the SPI tend to be equivalent to that of ESP. The PIT diagrams show that ensembles selected based on cumulative precipitations are not perfectly reliable, with observations too often falling below the forecast range in most catchments. Ensembles selected based on the SPI show a similar tendency, but in fewer catchments. In general, PIT values are closer to the diagonal when conditioning based on SPI values, especially with ESP_SPI3, which gives more reliable forecasts in most catchments.

Figures 4 and 5 have shown that the conditionings tend to increase sharpness and maintain or just slightly decrease reliability. Conditioning based on the SPI provides more consistent results between catchments and tends to produce more reliable forecasts. More specifically, conditioning based on SPI3 minimizes the loss in reliability and in overall performance comparatively to the ESP ensemble. In the following paragraphs, only HistQ_SPI3 and ESP_SPI3 were retained to further explore the quality of conditioned ensembles.

**3.2.2 Comparison of conditioned scenarios with the Sys4 base ensemble**

In Figure 6, we compare the quality of ESP, ESP_SPI3, HistQ and HistQ_SPI3 comparatively to Sys4. Figure 6 is similar to Figures 4 and 5 in that it represents the skill in overall performance, reliability and sharpness as a function of lead time, as well as the PIT diagrams at 45 days lead.

The behaviour of ESP is very similar to that of ESP_SPI3 with respect to Sys4. Both have better overall performance than

Sys4 for lead times shorter than 5 to 10 days, worse performance for lead times from 5 to 10 days and up to 20 days, and equivalent performance at longer lead times. In terms of reliability and sharpness, ESP and ESP_SPI3 are overall more reliable than Sys4 but not as sharp, though ESP_SPI3 becomes equivalent to Sys4 for lead times longer than 45 days. The PIT diagrams show that ESP and ESP_SPI3 are visually equivalent in terms of reliability, though the previously observed tendency of observations falling below the forecast range persists in a few catchments. This tendency may not be caused by

precipitation inputs but by the hydrological model.

If we now look at ensembles based on historical streamflows, we observe that HistQ performs worse than Sys4, at least for lead times shorter than 50 days. Even though HistQ is more reliable than Sys4, it is not as sharp, especially for lead times shorter than 30 days. HistQ_SPI3 also has lower overall performance than Sys4 but the gap in performance is reduced for lead times shorter than 15 days. HistQ_SPI3, following HistQ characteristics, provides forecasts that are more reliable than

Sys4, except at long lead times in some catchments. Contrary to HistQ, conditioning allows HistQ_SPI3 to be as sharp as Sys4 for horizons longer than 30 days. The reliability of HistQ and HistQ_SPI3 is confirmed by their PIT diagrams. These





diagrams also show that ensembles based on historical streamflows (HistQ) are more reliable than ensembles based on precipitation climatology (ESP).

### 3.2.3 Overall comparison of base and conditioned ensembles

The objective now is to see whether we succeeded in benefiting from the reliability of climatology and the sharpness of Sys4

when conditioning ensemble forecast scenarios. Figure 7 proposes a simultaneous evaluation of the reliability (PIT area) and sharpness (IQR) of ESP_SPI3 and HistQ_SPI3. For a given catchment, lead time and reference, the skill in reliability is plotted against the skill in sharpness. Each point corresponds to a catchment, each column corresponds to a lead time and each row corresponds to a forecast ensemble. Two references are chosen for each ensemble: ESP_SPI3 is evaluated against ESP and Sys4, and HistQ_SPI3, against HistQ and Sys4. Each reference is identified by its colour and shape (cf. legend). If a

point is located in the upper left part of the graph, the conditioned ensemble is more reliable but less sharp than the reference (indicated by the colour of the point) in the corresponding catchment. Reversely, if a point is located in the lower right part, the conditioned ensemble is sharper but less reliable than the reference. At best, both reliability and sharpness are improved, and points are located in the upper right part of the graph. At worst, both reliability and sharpness are deteriorated with respect to the reference, and points are located in the bottom left part of the graph.

Overall, we can observe that the conditioning tends to have more impact on reliability than on sharpness (y-axes extend further than x-axes). Also, conditioned ensembles are generally more reliable but less sharp than Sys4, and sharper but less reliable than the ensembles they are selected from. More specifically, we observe that:

-   For a lead time of 10 days, ESP_SPI3 and HistQ_SPI3 can be more reliable and sharper than the ensembles they are selected from. This applies to most catchments with ESP_SPI3, and to at least two catchments with HistQ_SPI3;

-   For a lead time of 30 days, fewer catchments benefit from a gain in both reliability and sharpness. The loss in sharpness and the gain in reliability with respect to Sys4 are less pronounced than for a lead time of 10 days. For instance, the maximum PITSS values for ESP_SPI3 move from 0.45 (for a lead time of 10 days) to 0.2 (for a lead time of 30 days) and those for HistQ_SPI3 move from 0.7 to 0.4. The gain in sharpness and the loss in reliability with regard to ESP and HistQ remain in the same ranges as observed for a lead time of 10 days;

-   For a lead time of 90 days, the gain of ESP_SPI3 over Sys4 is further reduced and varies with the catchment. The same is observed to a lesser extent for HistQ_SPI3, even though a positive impact of the conditioning on the reliability can still be observed in several catchments. At this lead time, both ESP_SPI3 and HistQ_SPI3 provide forecasts that are still sharper, yet less reliable, than the climatology they are selected from.

Figure 7 can also be interpreted in terms of distance between approaches. Indeed, the (0,0) coordinate corresponds to the

location of the references. From this perspective, we observe that ESP_SPI3 is closer to ESP than to Sys4 for a lead time of 10 days. But as the lead time increases, ESP_SPI3 becomes closer to Sys4 and further apart from ESP. The proximity between ESP_SPI3 and Sys4 at longer lead times can be attributed to the conditioning itself. The proximity between



ESP_SPI3 and ESP and their distance to Sys4 at shorter lead times may be attributed to the initialization of the climate model. Indeed, since initial hydrological conditions are the same for the three forecast ensembles, differences are caused by meteorological forcings only. The main difference between System 4 precipitations and climatology at such lead times is the initialization of the GCM, which leads to sharper System 4 forecasts in the first lead times. Similarly, we observe that

HistQ_SPI3 becomes closer to Sys4 as the lead time increases due to conditioning. However, its distance to HistQ remains the same at all lead times. This distance is probably due to the use of previous streamflow conditions as a conditioning criterion within HistQ. Therefore, the three ensembles, HistQ, HistQ_SPI3 and Sys4 are equally distant in the first lead times.

Table 3 proposes a ranking of the different ensembles investigated based on overall performance, reliability and sharpness

and for different lead time ranges: from 10 to 30 days, from 30 to 60 days and from 60 to 90 days. The rankings are based on the visual evaluation of Figure 5. The mean rank is calculated as the mean of the ranks obtained in the nine cells of the 3x3 table. Overall performance, reliability and sharpness are thus considered equivalent in this final ranking. Note that this may not be representative of operational expectations, since, in operational conditions, one could choose to emphasize one of the three characteristics over the others.

Based on Table 3, we can say that, if one seeks an overall performing ensemble with 10 to 30 days lead, one would use Sys4. For horizons longer than 30 days, ESP and ESP_SPI3 offer good alternatives. If one seeks, above all, a reliable ensemble, one could simply use HistQ, ESP, or even HistQ_SPI3 for lead times shorter than 30 days. However, for ensembles that are both sharp and reliable, and for horizons longer than 30 days, one could turn to the following ensembles: ESP_SPI3 for an emphasis on sharpness, or HistQ_SPI3 for an emphasis on reliability.

**3.3 Statistical evaluation of low flows**

We assess the performance of the ensemble forecast scenarios to forecast summer low flows and drought risks. Many ways of characterizing severe low flows and droughts exist in the literature (Mishra and Singh, 2010; Smakhtin, 2001; Tallaksen et al., 1997; WMO, 2008). In the following, the low-flow variables considered are the low-flow duration and deficit volume, both computed for the 80th exceedance percentile. In this section, only forecast horizons falling within the May to October

period are considered.

**3.3.1 Capacity of the ensembles to forecast low-flow events**

The capacity of the different systems to discriminate between low-flow events and non-events is assessed. Figure 8 presents the ranges of the Area Under the Curve (AUC) of the ROC diagram obtained from the five ensemble forecast scenarios, namely Sys4, ESP_SPI3, ESP, HistQ_SPI3 and HistQ. AUC values were assessed for the 80th exceedance percentile and for

lead times of 10 days, 30 days and 90 days. Each boxplot gathers the AUC values obtained in the 16 catchments. The letters below the boxplots result from the Friedman test (Lowry, 2008). This test consists in considering catchments as judges of the



five methods. The test, which is based on rankings as evaluated by the catchments, assesses whether the methods are significantly different by assessing whether their rankings resemble a random shuffling. Based on this test, two boxplots sharing a letter at a given lead time are not significantly different.

Results show that all ensembles but HistQ are very close in terms of discrimination. As expected, their performance

decreases as the lead time increases, except for HistQ, whose discrimination does not vary with the lead time. For all lead times, ESP significantly provides the best discrimination with most AUC values superior to 0.88. ESP_SPI3 and Sys4 are tied in terms of discrimination and appear as second best, with most AUC values greater than 0.82. HistQ_SPI3 is also very close to the performances of Sys4 and ESP_SPI3, but does not score as high as they do, especially for longer lead times. Nevertheless, HistQ_SPI3 mostly provides AUC values larger than 0.81. HistQ always provides AUC values between 0.8

and 0.9, except in Catchment 1, in which we have seen that this ensemble forecast has very low performances. Overall, ensembles based on hydrological modelling (Sys4, ESP and ESP_SPI3) provide the best skills in discrimination, at least for lead times shorter than 90 days, probably because they take into account initial hydrological conditions. We note that all these conclusions are also valid when the 60th exceedance percentile is used as threshold (not shown).

### 3.3.2 Capacity of the ensembles to forecast low-flow variables

We now compare the forecast systems based on variables of interest for water management during low flows, namely the weekly deficit duration and the weekly deficit volume. The weekly deficit duration corresponds to the number of days per week during which the daily streamflow is below a given threshold. The weekly deficit volume corresponds to the flow volume per week below this threshold. Figure 9 presents the PIT areas obtained with Sys4, ESP_SPI3, ESP, HistQ_SPI3 and HistQ when forecasting the weekly number of days below the 80th exceedance percentile. Boxplots represent the range of

PIT areas obtained over the catchment set. Results are presented for lead times of two weeks, five weeks and twelve weeks (columns). Again, letters represent the results of the Friedman test. Two boxplots that share a letter are not significantly different. Figure 10 proposes the same evaluation for the weekly streamflow deficit volume below the 80th exceedance percentile.

Figure 9 shows that the difference between the five ensembles is very tenuous when forecasting the deficit duration. For

instance, all lower and upper quartiles of Sys4, ESP_SPI3, ESP and HistQ_SPI3 are included in the [0.01, 0.08] interval of PIT area values, regardless of the lead time. Overall, ESP, ESP_SPI3 or HistQ_SPI3 perform best to forecast the deficit duration. All ensembles but HistQ provide quite reliable forecasts (PIT area values close to zero). HistQ_SPI3 is significantly the best performing ensemble for a lead time of two weeks. For a lead time of five or twelve weeks, both ESP and HistQ_SPI3 are the best options. The analysis of the corresponding PIT diagrams (not presented) showed that all

ensembles are equivalently reliable, except for HistQ, which systematically overestimates the deficit duration.

The gap between ensembles widens when looking at the deficit volume (Figure 10). For lead times of two and five weeks, ESP and ESP_SPI3 provide consistently reliable ensembles, and lower PIT areas than the others. For a lead time of twelve





weeks, ESP_SPI3, along with Sys4 and HistQ_SPI3, provide the most reliable ensembles. The corresponding PIT diagrams (not presented) showed that HistQ_SPI3 tends to underestimate durations at all lead times. Ensembles issued with hydrological modelling also slightly underestimate the deficit volume at long lead times. Overall, ESP_SPI3 systematically appears to be one of the best options to forecast deficit volumes.

**3.4 Drought impact evaluation**

Figure 11 illustrates the case of the 2003 drought with the streamflow forecasts issued on July $1^{st}$ 2003 for the three months ahead. The figure focuses on catchment 5, the Azergues at Lozanne, in which the 2003 drought was hydrologically more severe than the reference 1976 drought. Each column represents the graphs obtained with one of the five ensemble forecasts (Sys4, ESP_SPI3, ESP, HistQ_SPI3 and HistQ). The upper row presents the graphical representation we propose to assess drought risks based on the ensemble forecasts. The graphs represent the deficit duration against the deficit volume, both computed based on the $80^{th}$ exceedance percentile. The graph is divided into 49 boxes corresponding to possible combinations and ranges of deficit volumes and durations. The colour within each of these boxes indicates the percentage of ensemble members that falls within each box. Coloured dots represent the observation and two references: the 1976 drought and the historical mean duration and deficit volume over the forecast period (climatology). The lower row presents the corresponding hydrographs over the forecast period. The black line represents the observed streamflow, the red line represents the $80^{th}$ exceedance percentile and the blue lines represent the members of the ensemble forecast.

All ensembles produce similar patterns, but with different probabilities. The maximum probability is obtained with HistQ_SPI3 with 60 % of the ensemble members falling in the same cell. Ensembles based on hydrological modelling reach maximum probabilities of 20 to 30 %, and HistQ does not exceed a probability of 14 %. These colours translate in a way the sharpness of the ensemble forecasts. The objective with the graph is to have a maximum of darker cells close to the observation (represented by the black dot). We observe that the graph obtained with HistQ puts equivalent weights to a wide range of scenarios indicating no risk to high risks. This ensemble thus conveys little information to assess drought risks. HistQ_SPI3, as opposed to HistQ, offers a more confident risk assessment with the highest forecast probabilities and only three coloured cells. Eighty percent of the forecast members indicate a drought equivalent or more severe than that of 1976. The high probability may be explained by the fact that SPI forecast members and initial hydrological conditions were often best represented by the same driest year (as suggested by the hydrographs), namely 1976.

The ESP forecast provides a wider view of risks, with higher probabilities located in the upper right part of the graph, and small probabilities of having months with moderately dry conditions. ESP is able to forecast a more severe event than observed during the 1976 drought. This good performance can only be attributed to the initial hydrological conditions since ESP does not have any information on future precipitations apart from climatology. Conditioning ESP (ESP_SPI3) slightly reduces the number of coloured cells with slightly higher probabilities in some of the upper right cells. The difference



between ESP and ESP_SPI3 is clear when looking at the hydrographs. With ESP_SPI3, the number of high-flow peaks is reduced.

Sys4 also provides a quite good risk assessment since only upper right cells are coloured. While ensembles based on hydrological modelling, i.e. ESP, ESP_SPI3 and Sys4, are limited by the capacity of the model to reproduce small low-flow

variations and thus slightly underestimate the deficit volume, ensembles based on historical streamflows are limited within the range of past precipitation and streamflow scenarios. This highlights the fact that the studied methods, and here specifically Sys4, ESP_SPI3 and HistQ_SPI3, have different limitations, but also different assets. We have illustrated their performances to forecast a given drought event in France. We should however keep in mind that different contexts might penalize or favour different methods.

**4 Conclusion**

We have investigated the potential of seasonal streamflow forecast ensembles built by conditioning precipitation climatology and historical streamflows based on precipitation indices derived from ECMWF System 4 (GCM) forecasts. In a first step, the performance of the conditioned ensembles was assessed in terms of overall performance, sharpness and reliability for lead times up to 90 days. Here are the main conclusions from this comparison:

- Selecting traces within precipitation climatology or historical streamflow generally improved sharpness and decreased reliability. Conditioning based on the SPI provided more consistent results between catchments and more reliable forecasts than conditioning based on cumulative precipitations. More specifically, conditioning based on SPI3 improved overall performance as compared to historical streamflow and maintained overall performance as compared to precipitation climatology used as input to a hydrological model, while providing reliable forecasts.

- A simultaneous evaluation of the sharpness and reliability of the conditioned ensembles showed that conditioning led to ensembles that were more reliable and less sharp than streamflow forecasts generated from System 4 precipitations, and less reliable and sharper than the ensembles they were selected from. Also, the conditioned ensembles benefit from the information of either precipitation climatology or historical streamflows at shorter lead times and from the information of GCM-based forecasts at longer lead times.

- Ensembles selected from precipitation climatology and historical streamflow offer a good compromise between sharpness and reliability, with an emphasis on sharpness with precipitation climatology, and an emphasis on reliability with historical streamflows.

The performance of the ensembles in forecasting low-flow events and low-flow variables was then evaluated, with an illustration on the 2003 drought in France. Their capacity to discriminate between low-flow events and non-events and their

capacity to forecast streamflow deficit volume and duration, as defined by the $80^{th}$ exceedance percentile, were assessed. The main conclusions from this second evaluation are:





- Forecast ensembles using hydrological modelling provided better discrimination than ensembles based on historical streamflows. Nevertheless, all forecast ensembles provided good performance, except for historical streamflows for lead times shorter than a month.

- Even though differences between ensembles are tenuous when forecasting low-flow duration, the gap widens when forecasting deficit volume. The ensemble selected within precipitation climatology systematically provides some of the most reliable deficit volume forecasts.

- Lastly, a graphic representation of the forecast drought risks was proposed. It was illustrated with the 2003 drought. We showed that, for this drought event, conditioned ensemble forecasts (either based on precipitation climatology or historical streamflows) provided good drought risk assessment.

We investigated conditionings within climatology solely based on past precipitations and catchment conditions. SPI values were computed after an aggregation of System 4 precipitation forecasts at the catchment scale, therefore the conditioning and the spatial aggregation were independent. Further investigations could assess the potential of this method for spatial downscaling of System 4 precipitation forecasts.

In this paper, the conditioning based on the forecast SPI or cumulative precipitations for the three coming months puts an equivalent weight on all three lead times to select past precipitations. As we showed in this paper, the System 4 forecasts have more skill for the coming month than for the second and third months. Therefore, we could explore a weighting of these three forecast lead times, to put more weight on the first month lead time in the selection of past precipitations.

One important parameter to forecast low flows and droughts is the temperature. A more advanced approach would consist in selecting past scenarios based on the SPEI (Standardized Precipitation-Evapotranspiration Index) calculated from seasonal precipitation and temperature forecasts.

Finally, other types of combinations can be found in the literature and could be investigated along with the proposed conditionings. As an example, Werner et al. (2005) or Shukla et al. (2012) have investigated the use of medium-range weather forecasts to improve long-range forecasting. These approaches are based on the fact that short-term events are well forecast by short-term to medium-term forecasts issued by GCMs and that the benefit from medium-range forecasts can be extended to longer lead times through the inertia of a catchment.

### Acknowledgements

The first author was partly funded by the DROP project (Benefit of governance in DROught adaptation) of the Interreg IVB NWE programme of the European Union. The second and the third authors were partly funded by the IMPREX project supported by the European Commission under the Horizon 2020 Framework programme, with grant nr 641811. The authors thank Météo-France and SCHAPI for providing climate and hydrological data respectively, and ECMWF for providing seasonal forecast data and for hosting the first author for two weeks.





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





**Figure 1** Location in France and hydrological regime of the 16 catchments. Solid lines represent mean interannual monthly flows. Grey-shaded areas represent the 10th and 90th percentiles of interannual monthly flows. Dotted red lines represent the 80th exceedance percentile (i.e. the daily flow exceeded by 80 % of the data). The catchments are numbered from the smallest to the largest. Statistics are computed over the streamflow record available for each catchment, i.e. 36 to 52 years (cf. Table 1).





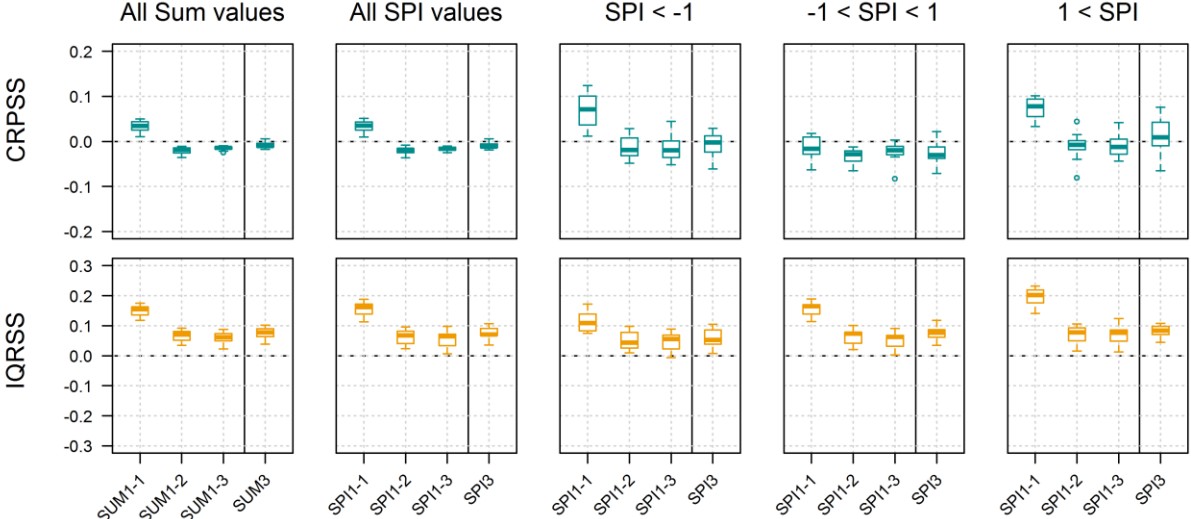

**Figure 2** CRPSS and IQRSS of SPI forecasts and forecasts of cumulative precipitations produced from bias corrected System 4 precipitation forecasts. The reference for the skill scores is climatology. Skill scores are presented for statistics calculated over one month and three months altogether (Sum1 and Sum3; SPI1 and SPI3). Columns correspond to scores computed for sums, SPI values, SPI values smaller than -1 (dry), SPI values within -1 and 1 (normal) and SPI values greater than 1 (wet).

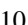

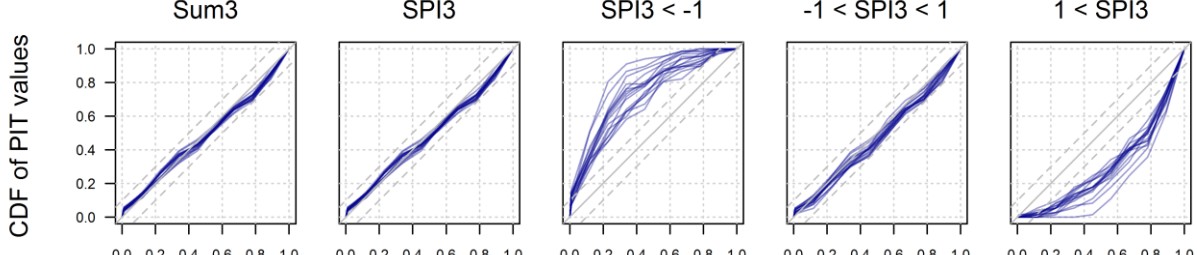

**Figure 3** Reliability of SPI forecasts and forecasts of cumulative precipitations produced from bias corrected System 4 precipitation forecasts. PIT diagrams are presented for statistics calculated over the first three months altogether (Sum3 and SPI3). Columns correspond to scores computed for sums, SPI values, SPI values smaller than -1 (dry), SPI values within -1 and 1 (normal) and SPI values greater than 1 (wet).






**Figure 4** Skill scores (CRPSS, IQRSS, PITSS; first three rows) and PIT diagrams for a lead time of 45 days (last row) of the conditioned ensemble forecast scenarios: HistQ_Sum3, HistQ_Sum1, HistQ_SPI3 and HistQ_SPI1. In the skill scores, the reference forecast is the base ensemble HistQ. Each line represents one of the 16 catchments investigated.





**Figure 5** Same as Figure 4 but the forecast ensembles are ESP_Sum3, ESP_Sum1, ESP_SPI3 and ESP_SPI1 and the reference for the computation of the skill is ESP.





**Figure 6** Same as Figure 3 but the forecast ensembles are ESP, ESP_SPI3, HistQ and HistQ_SPI3 and the reference for the computation of the skill is Sys4.





**Figure 7** PITSS (reliability) versus IQRSS (sharpness) for ESP_SPI3 (upper row) and HistQ_SPI3 (lower row), and lead times of 10, 30 and 90 days (columns). ESP_SPI3 is compared to Sys4 (red) and ESP (grey), while HistQ_SPI3 is compared to Sys4 (red) and HistQ (blue). Each point represents one of the 16 catchments.




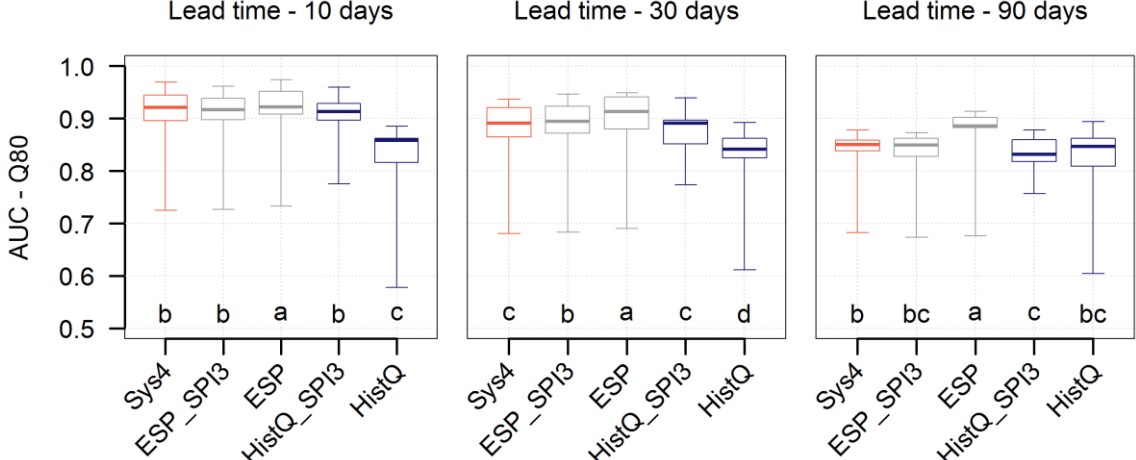

**Figure 8** Ranges of the Area Under the Curve (AUC) of the ROC diagram based on the 80[th] exceedance percentile for each of the five selected ensemble forecasts (Sys4, ESP, HistQ, ESP_SPI3, HistQ_SPI3). Boxplots gather the AUC values for the 16 catchments. The boxes extend to the 25[th] and 75[th] percentiles and the whiskers, to the data extremes. Graphs are presented for 10-day, 30-days and 90-day lead times (columns). The letters below the boxplots result from the Friedman test. For a given lead time, two boxplots sharing a letter are not significantly different.





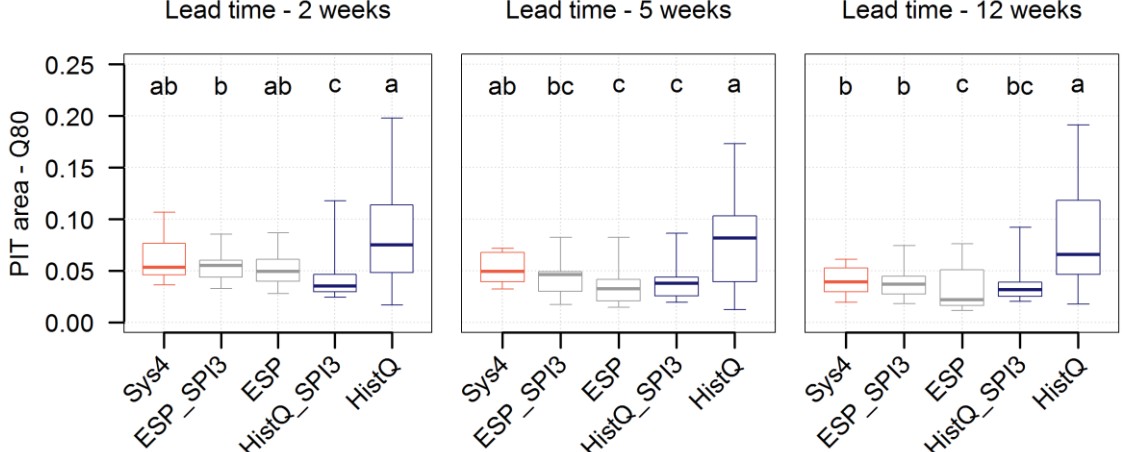

**Figure 9** Same as Figure 7 but for PIT area ranges computed for deficit duration. Ranges are represented by boxplots which gather the PIT areas for the 16 catchments. Graphs are presented for lead times of two weeks, five weeks and twelve weeks (columns).




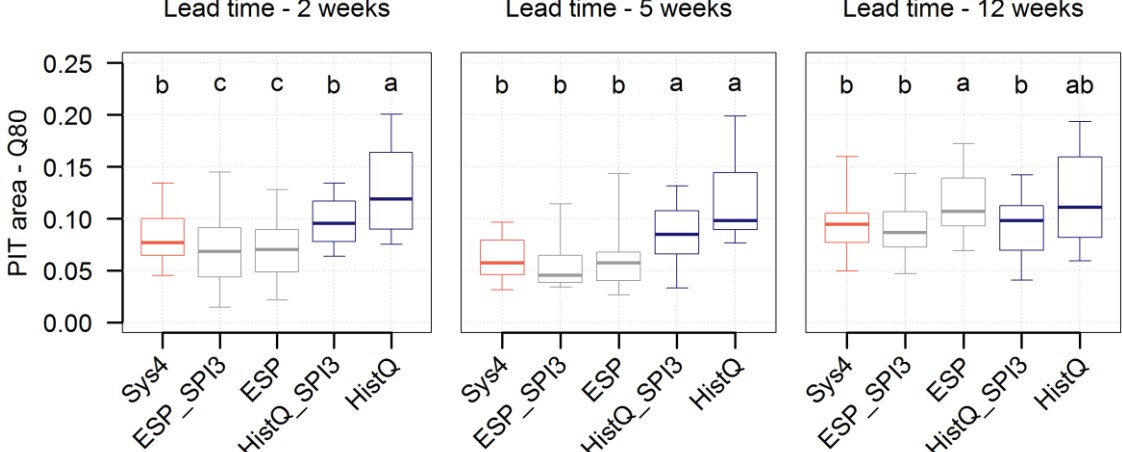

**Figure 10** Same as Figure 8 for deficit volume.




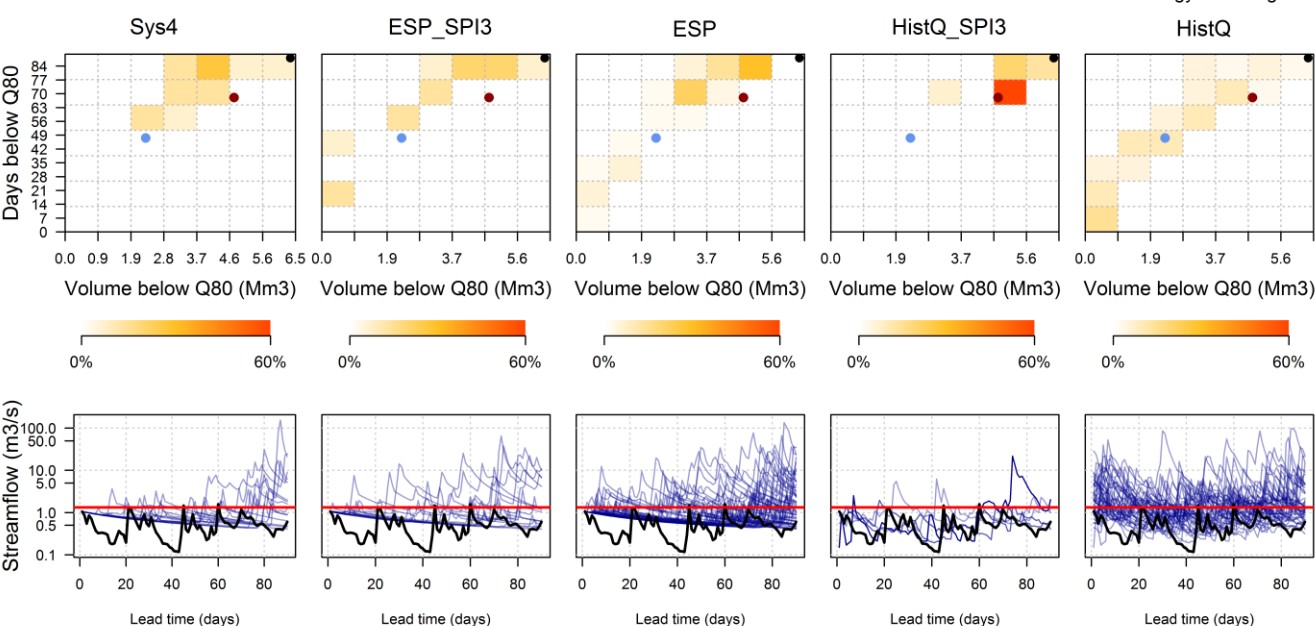

**Figure 11** Risk graphs presenting the probabilities of deficit duration versus deficit volume based on the 80th exceedance percentile (upper row) and corresponding hydrographs (lower row). The maximum probability varies with the ensemble and the situation and is indicated in the colour scale. The black point corresponds to the observation, the dark red dot to the drought of 1976 and the blue dot to the mean duration and deficit volume observed in past streamflows. Each column corresponds to one of the five ensemble forecasts. Forecasts were issued for the Azergues at Lozanne (catchment 5) for the period running from 1 July to 30 September 2003.




**Table 1** River and gauging station, period with available streamflow observations, area and main hydroclimatic characteristics of the 16 catchments (ranked from the smallest to the largest). The mean annual streamflow is computed over the period of streamflow availability. The mean annual precipitation and evapotranspiration are computed over the 1958-2010 period.

| # | River | Gauging station | Streamflow availability | Area (km²) | Mean annual precipitation (mm/yr) | Mean annual potential evapotranspiration (mm/yr) | Mean annual streamflow (mm/yr) |
|---|---|---|---|---|---|---|---|
| 1 | Andelle | Vascoeuil | 01/01/1973 - 27/02/2010 | 377 | 952 | 628 | 332 |
| 2 | Orne Saosnoise | Montbizot *[Moulin Neuf Cidrerie]* | 01/12/1967 - 04/03/2010 | 501 | 735 | 696 | 163 |
| 3 | Briance | Condat-sur-Vienne *[Chambon Veyrinas]* | 01/01/1966 - 28/03/2010 | 605 | 1100 | 706 | 427 |
| 4 | Ill | Didenheim | 01/11/1973 - 02/03/2010 | 668 | 956 | 664 | 309 |
| 5 | Azergues | Lozanne | 01/01/1965 - 28/03/2010 | 798 | 931 | 689 | 296 |
| 6 | Seiche | Bruz *[Carcé]* | 01/12/1967 - 11/03/2010 | 809 | 732 | 696 | 181 |
| 7 | Petite Creuse | Fresselines *[Puy Rageaud]* | 01/08/1958 - 28/03/2010 | 853 | 899 | 680 | 316 |
| 8 | Sèvre Nantaise | Tiffauges *[la Moulinette]* | 01/11/1967 - 04/03/2010 | 872 | 898 | 712 | 331 |
| 9 | Vire | Saint-Lô *[Moulin des Rondelles]* | 01/01/1971 - 03/02/2010 | 882 | 958 | 629 | 448 |
| 10 | Orge | Morsang-sur-Orge | 01/10/1967 - 07/03/2010 | 934 | 658 | 680 | 131 |
| 11 | Serein | Chablis | 01/08/1958 - 03/03/2010 | 1119 | 842 | 675 | 220 |
| 12 | Sauldres | Salbris *[Valaudran]* | 01/01/1971 - 28/03/2010 | 1220 | 803 | 684 | 240 |
| 13 | Eyre | Salle | 01/01/1967 - 19/03/2010 | 1678 | 1025 | 785 | 323 |
| 14 | Arroux | Etang-sur-Arroux *[Pont du Tacot]* | 01/11/1971 - 27/03/2010 | 1792 | 981 | 655 | 390 |
| 15 | Meuse | Saint-Mihiel | 01/07/1968 - 03/01/2010 | 2543 | 948 | 639 | 372 |
| 16 | Oise | Sempigny | 01/08/1958 - 02/03/2010 | 4320 | 805 | 639 | 250 |





**Table 2** Summary of the methodology used to build the ensemble forecast scenarios.

| Name | Statistic on seasonal forecast used as condition | Additional condition | Size | Initial hydrological conditions | Hydrological model | Precipitation forecast |
|---|---|---|---|---|---|---|
| **HistQ** | No condition | - | Between 35 and 51 depending on flow data availability (see Table 1) | no | no | no |
| **HistQ_Sum3** | Precipitation volume | previous streamflow | 15 or 51 | yes | no | no |
| **HistQ_Sum1** | Monthly precipitation volume | | | yes | no | no |
| **HistQ_SPI3** | SPI3 | | | yes | no | no |
| **HistQ_SPI1** | SPI1 | | | yes | no | no |
| **ESP** | No condition | - | 50 | yes | yes | no |
| **ESP_Sum3** | Precipitation volume | - | 15 or 51 | yes | yes | no |
| **ESP_Sum1** | Monthly precipitation volume | | | yes | yes | no |
| **ESP_SPI3** | SPI3 | | | yes | yes | no |
| **ESP_SPI1** | SPI1 | | | yes | yes | no |
| **Sys4** | No condition | - | 15 or 51 | yes | yes | yes |



**Table 3** Rankings of the Sys4, ESP_SPI3, ESP, HistQ_SPI3 and HistQ streamflow ensembles, as evaluated by three evaluation criteria (in rows) and three lead time ranges (columns). The mean rank is calculated for each ensemble and is a simple mean of the ranks obtained by this ensemble in the nine cells of the 3x3 table.

|  | **10-30 days lead** | **30-60 days lead** | **60-90 days lead** |
|---|---|---|---|
| **Overall performance** | 1. Sys4<br>2. ESP_SPI3<br>2. ESP<br>4. HistQ_SPI3<br>5. HistQ | 1. Sys4<br>1. ESP_SPI3<br>1. ESP<br>4. HistQ_SPI3<br>5. HistQ | 1. Sys4<br>1. ESP_SPI3<br>1. ESP<br>4. HistQ_SPI3<br>4. HistQ |
| **Sharpness** | 1. Sys4<br>2. ESP_SPI3<br>3. ESP<br>4. HistQ_SPI3<br>5. HistQ | 1. Sys4<br>2. ESP_SPI3<br>3. HistQ_SPI3<br>4. ESP<br>5. HistQ | 1. Sys4<br>1. ESP_SPI3<br>1. HistQ_SPI3<br>4. ESP<br>5. HistQ |
| **Reliability** | 1. HistQ<br>2. HistQ_SPI3<br>3. ESP<br>4. ESP_SPI3<br>5. Sys4 | 1. HistQ<br>2. ESP<br>3. HistQ_SPI3<br>4. ESP_SPI3<br>5. Sys4 | 1. HistQ<br>2. ESP<br>3. HistQ_SPI3<br>4. ESP_SPI3<br>4. Sys4 |
| **Mean Rank** | **Sys4 : 2.22** | **ESP_SPI3 : 2.33** | **ESP : 2.44**  **HistQ_SPI3 : 3.11**  **HistQ : 3.55** |

