# Peer review of "Seasonal streamflow forecasting by conditioning climatology with precipitation indices"

_Hydrology and Earth System Sciences, 2016_

## Referee Comment (RC1) · Anonymous Referee #1 · 24 Jul 2016

As an outsider to the professional academic world, I feel that I cannot speak with unquestionable credibility to the novelty or scientific soundness of this manuscript – I am simply not familiar enough with the wealth of recent research into seasonal hydrologic forecasting. However, I can supply my overall impression of this work, which may be useful given my background in operational hydrologic forecasting.

The authors reference several studies that utilized approaches similar to the one undertaken here – conditioning historical observation-based ensembles to improve forecasts generated from these ensembles. Thus, the fundamental direction of the current study is not overly original. However, the manner in which the conditioning was applied – using GCM- and climatology-derived precipitation indices to select the most relevant historical ensembles – does appear to be a novel approach.

[Figure]

The potential utility of this approach is presented well in Figure 2, where the precipitation indices generated from the GCM hindcasts (ECMWF Sys4) are compared against those generated from the historical observations. As the authors state, the Sys4 indices perform at least as well as the base indices overall (CRPSS), even outperform at one month lead time, but are consistently sharper (IQRSS). Further, the Sys4 indices have good reliability overall (Figure 3). The reliability of the indices falters when looking at only drier than normal or only wetter than normal conditions, but this seems to be unavoidable with any forecasting approach.

Despite the prefaced potential of using the Sys4 precipitation indices to condition, or subset, historical ensembles, this study's results offer just marginal practical insight:

1) Subsetting the ensembles based on the precipitation indices improve the HistQ performance more than the it does the ESP performance. This result is not very useful, however, since the HistQ approach is rudimentary (and likely rarely used), and the primary benefit of the conditioning is seen during short lead times (which is simply the effect of blending from the last streamflow observation).

2) For ESP, SPI-conditioning appears to outperform SUM-conditioning, but this statement is qualitative at best and neither set of conditioned ensembles provides any notable improvements over the base ensembles. Compared to the base ESP ensembles, the sharpness of the ESP_SPI3 ensembles was improved by up to 10% but the reliability was degraded by up to 40% (Figure 7).

3) The conditioning improved the performance of HistQ ensembles in forecasting low flow events and variables, but the conditioned ensembles were still less skillful than the Sys4 and ESP/ESP_SPI3 ensembles.

4) The authors state that the ESP_SPI3 approach "systematically appears to be one of the best options to forecast deficit volumes." However, this conclusion is very subjective, as it is not authoritatively substantiated by the results presented in Figures 9 and 10.

[Figure]

Although several pages of this manuscript are spent discussing the results in great detail, and the authors walk through the discussion in a relatively clean, scientific manner, much of this discussion is centered around tangential topics. For example, the comparisons between the conditioned ESP/HistQ ensembles to the Sys4 ensembles seem irrelevant given that the conditioning did little to improve, and actually degraded in some cases, the skill compared to the base ensembles. Thus, comparing the conditioned ensembles to the Sys4 ensembles is equivalent to comparing the base ensembles to Sys4, which of course is unnecessary. The results should be restricted to and presented with the stated goal of the study in mind – improving the skill of historical observation-based ensemble forecasting systems.

Unfortunately, because there is little to report on the utility of applying this conditioning method to seasonal streamflow and low flow forecasting, the authors may need to redesign and/or include other experiments before resubmitting this paper. One suggestion, actually offered by the authors, is to examine the utility of using SPEI to condition the ensembles. Although the SPI is likely sufficient to appropriately subset historical precipitation ensembles, it may not be sufficient from a streamflow perspective. It seems likely that the relative magnitude of an individual SPI value may not always be translated into a similar relative magnitude flow or volume value if ET is a major hydrologic control in the watershed of interest (i.e. late season streamflows can be very different following extended dry but mild vs extended dry but hot conditions). Thus, conditioning the ensembles with both precipitation- and temperature-driven indices may provide more robust results.

Lastly, the underlying standard of this manuscript is the stated inherent reliability of historical observation-based ensembles, but this is a bit misleading. In true forecasting (not hindcasting), climatology-driven predictions may not be all that reliable. Several decades worth of historical information is often sought to build an ensemble forecasting system, but the climatic regime of the forecast area may be changing too rapidly for this. Thus, the distribution functions of actual forecasts and their corresponding observations may be offset from one another (i.e. not fall on a 1:1 line). Perhaps the authors should frame the goal more along the lines of using the conditioning to sharpen the ensembles, and less along the lines of marrying the reliability of historical ensembles with the sharpness of GCMs.

---

## Referee Comment (RC2) · Anonymous Referee #2 · 26 Jul 2016

This study proposes an approach to improve short- and long-range (10-90 days) streamflow forecasts by conditioning resampled historical observations based on ECMWF System 4 forecasts. The conditioning is applied on both precipitation and streamflow records. Results are compared with historical resampled streamflow and ensemble streamflow prediction (ESP) as reference forecasts. Overall, the paper is well written and provides good assessments of different model performances. Nevertheless, I am concerned with the proposed method to improve streamflow forecasts (selection of resampled data based on GCM forecasts) as well as the results (week performance of the proposed method). Therefore, I think the paper is not ready for publication and requires major revision.

Major comments:

[Figure]

1) The manuscript states that (P4, L9) the aim of this study is "to generate forecasts that benefit from the reliability of climatology-based ensembles and the sharpness of System 4 precipitation forecasts." First the proposed method does not seem to benefit from the sharpness of System 4, rather the reason for increased precision (sharpness) in the conditioned forecasts is due to the reduced ensemble size which is independent of the System 4's degree of uncertainty. Second, the results (e.g. Figures 4-5) show that except for some marginal improvements in forecasts for short lead times (Figure 4 upper row), the proposed method degrade the performance of the reference methods (CRPSS and PITSS are negative). In several instances in the manuscript (such as P9, L17) the authors discuss the improvements to the sharpness of the forecasts using their conditioning approach while reliability and performance have declined compared to the reference methods which undermines the sharpness improvements. The authors state that "…the PIT diagrams at 45 days show that this decrease does not affect the overall reliability of the conditioned ensembles" This again shows that the proposed method has not been able to improve upon the conventional approaches.

2) The proposed method selects forecast ensemble members based on their closeness to some statistics (P8, L17). The procedure to choose the number of ensemble members to keep, however, is not explained. Is the number of selected runs subjectively chosen? If so a sensitivity analysis needs to be conducted.

3) The method conditions the resampled precipitation and streamflow data to GCM forecasts. However, GCM forecasts are uncertain particularly at seasonal scales. That might explain why the overall results do not show improvements compared with conventional ESP. In particular, authors need to discuss how the method will perform in regions with high topographical variations (considering that the low-resolution GCMs cannot capture the regional hydroclimatic variations). Related to this please discuss why you compare the proposed conditioning approach (based on SYS4) with results of SYS4?

4) Please clarify are the statistics (section 2.4.2) calculated for each ECMWF ensemble

member separately or for the average of the 51 ensemble runs?

5) P8, L25: "when directly selecting scenarios from past streamflow observations, the last observed streamflow is added as a conditioning criterion in the computation of the Euclidian distance." This is problematic as the last observed (previous year's(?)) streamflow is not a good indicator of the next year's streamflow in particular with regard to high and low flows which are driven by several hydroclimatic factors that do not necessarily repeat at consecutive years.

6) Resampled precipitation is considered to drive the hydrologic model, however, the mean interannual potential evapotranspiration is used instead of the resampled one. Considering that PET might have a substantial role in low flow forecasts, I recommend using the resampled PET as well.

7) P12, L12: "The rankings are based on the visual evaluation of Figure 5." Visual evaluation is not an appropriate ranking approach.

8) Results of section 3.4 are based on only one drought event for one catchment and cannot provide sufficient evidence for the overall performance of the methods.

9) P6, section 2.3.1 Please elaborate further on the differences between CRPS and PIT and how they should be interpreted when they show inconsistent results (e.g. Fig 4).

10) Multi-model averaging methods (such as simple mean, Bayesian Model Averaging (BMA) etc.) (Duan et al. 2005, Najafi et al. 2015, Raftery et al. 2005) have shown to improve short and long term hydrologic forecasts. I would suggest discussing the application of these approaches to merge the ensemble of forecasts obtained from different methods in this study.

Specific comments:

- Abstract "...forecasts based on GCM outputs can offer sharper ensembles...": does "sharper" refer to more precise? Related to this please define "sharpness" and "reliability" before using these terms, in the Introduction. - L15: ECMWF System 4: Please expand the full name. - Abstract: "The four conditioned precipitation scenarios were used as input to the GR6J hydrological model to obtain eight conditioned streamflow forecast scenarios": The statement is vague as to how four precipitation scenarios result in eight streamflow scenarios? - P2, L19: ESP is one of the streamflow forecast methods which need to be discussed here. Also please note that in ESP all historical meteorological forcings can be resampled to run the hydrological model (not just precipitation as stated in LP2, L27) - P4, L3 Statement is not clear "although the ensemble conditioned from historical streamflows, which was the. . ." - P4, L12-15: Please move to the results section. - P4, L17: Please define "discrimination" - P5, L3: Please explain how many grid cells lie within each catchment in average. How was the aggregation performed? Please also indicate the forecast starting date. - P5, L23: What do you mean by "systematically"? - P5, L31-33: What is the range of KGE values? Please show the equations for KGE and 1-bias and include their ranges. - P6, L9: Please change "The CRPS averages over the evaluation period the area between the cumulative forecast distribution. . ." to "The CRPS averages the area between the cumulative forecast distribution. . . over the evaluation period." Similarly, for L12. - P7, L3: What is the "reference"? Is it HisQ? Please define. - I suggest bringing section 2.4 before section 2.3. - Figure 2: What is the difference between SUM1-3 and SUM3 - P9, L1 "The reference forecast used to compute the skill scores is historical precipitations (i.e. climatology)": Do you mean hydrologic model simulation driven by historical precipitation? - P9, L3 "SPI forecasts issued from System 4 are reliable overall and in standard precipitation conditions" please provide a reference

Raftery, Adrian E., et al. "Using Bayesian model averaging to calibrate forecast ensembles." Monthly Weather Review 133.5 (2005): 1155-1174. Najafi, M. and Moradkhani, H. (2015). "Ensemble Combination of Seasonal Streamflow Forecasts." J. Hydrol. Eng., 10.1061/(ASCE)HE.1943-5584.0001250, 04015043. Duan, Qingyun, et al. "Multi-model ensemble hydrologic prediction using Bayesian model averaging." Advances in Water Resources 30.5 (2007): 1371-1386.

---

## Author Comment (AC1) · 19 Sep 2016

**Response to Reviewer#1**

The authors want to thank Reviewer#1 for the valuable comments on our manuscript. We provide below our answers to the comments.

**Reviewer 1**

As an outsider to the professional academic world, I feel that I cannot speak with unquestionable credibility to the novelty or scientific soundness of this manuscript – I am simply not familiar enough with the wealth of recent research into seasonal hydrologic forecasting. However, I can supply my overall impression of this work, which may be useful given my background in operational hydrologic forecasting.

**Reviewer's comment (RC):** The authors reference several studies that utilized approaches similar to the one undertaken here – conditioning historical observation-based ensembles to improve forecasts generated from these ensembles. Thus, the fundamental direction of the current study is not overly original. However, the manner in which the conditioning was applied – using GCM- and climatology-derived precipitation indices to select the most relevant historical ensembles – does appear to be a novel approach.

Authors' reply (AR): Our study is clearly not the only one in the topic. As we have mentioned on Page 3, lines 2-3, the conditioning or weighting of past observations based on climate signals is a recent topic of research, and several studies are emerging to investigate the best approaches to be used in order to extend the skill of seasonal hydrometeorological predictions. In this regard, our study aims to contribute to this research area and the approach and application that we propose in the paper is original. Studies that use GCM-derived precipitation indices as conditioning indices are rare and applications of conditioning approaches over mid-latitudes (in our case, France), where the reliability of seasonal weather predictions is, in general, low, are important contributions to the community. In addition, in this study:
- this specific conditioning method was applied simultaneously to a set of sixteen catchments, which, to our knowledge, had not been done previously,
- we distinguish between performance associated with sharpness and reliability,
- the latest GCM forecasts in date for Europe (System 4 from ECMWF) were used. This is of importance since seasonal forecasting in meteorological centres has been improving in the past decade, and to our knowledge, no other study tried this conditioning approach as extensively as in our study with these latest forecasts available.

We propose to add a sentence at the beginning of Section 1.3 (Scope of the study) to highlight these points and the originality of our paper in the light of the existing literature.

**RC:** The potential utility of this approach is presented well in Figure 2, where the precipitation indices generated from the GCM hindcasts (ECMWF Sys4) are compared against those generated from the historical observations. As the authors state, the Sys4 indices perform at least as well as the base indices overall (CRPSS), even outperform at one month lead time, but are consistently sharper (IQRSS). Further, the Sys4 indices have good reliability overall (Figure 3). The reliability of the indices falters when looking at only drier than normal or only wetter than normal conditions, but this seems to be unavoidable with any forecasting approach.
Despite the prefaced potential of using the Sys4 precipitation indices to condition, or subset, historical ensembles, this study's results offer just marginal practical insight:

1) Subsetting the ensembles based on the precipitation indices improve the HistQ performance more than it does the ESP performance. This result is not very useful, however, since the HistQ approach is rudimentary (and likely rarely used), and the primary benefit of the conditioning is seen during short lead times (which is simply the effect of blending from the last streamflow observation).

AR: The reviewer has clearly understood the aims of the analysis illustrated in Figure 2 and we acknowledge the positive comment provided. We believe this first step in analysing the performance of the precipitation-based conditioning indices is essential prior to analysing the conditioned outputs in terms of streamflow, given the non-linearity in the transformation of precipitation into streamflow in a hydrological model. We also included HistQ in our study because this is a "poorman's approach" that can serve as a naïve benchmark, where no hydrological model but only a long streamflow time series of records is available. One of the objectives of our study was to see whether this rudimentary approach could be turned into a valuable one provided that precipitation anomalies are available. While we agree that the improvement in the first days/weeks can be due to the assimilation of the last streamflow observation, the effect of this data assimilation technique usually decreases with lead time. The improvement observed in sharpness, for instance, is then mostly due to the conditioning. This improvement is one of the aims of several operational seasonal hydrological prediction systems: obtain sharper predictions, while maintaining reliability.
* * *
RC: 2) For ESP, SPI-conditioning appears to outperform SUM-conditioning, but this statement is qualitative at best and neither set of conditioned ensembles provides any notable improvements over the base ensembles. Compared to the base ESP ensembles, the sharpness of the ESP_SPI3 ensembles was improved by up to 10% but the reliability was degraded by up to 40% (Figure 7).

AR: The comparison between SPI-conditioning and SUM-conditioning over ESP is illustrated in Figure 5. From this figure, we can see that conditioning on SPI (third and fourth columns) provides better scores over (or at least do not degrade the score of) the reference (base ESP ensembles) than conditioning with SUM (first and second columns). For instance, based on the IQRSS results, we can see that the SUM-conditioning may decrease sharpness in some cases, whereas the SPI-conditioning guarantees to maintain or even increase sharpness. Based on the PIT diagram analysis, the SUM-conditioning causes an overprediction of observations, whereas the SPI-conditioning clearly limits this effect. Figure 7, mentioned by the reviewer, provides the means for a more quantitative analysis. However, it is restricted to the results for SPI-conditioning, since this one was already qualitatively better than the SUM-conditioning from the analysis in Figure 5. Figure 7 shows that we can lose in reliability (PIT area) in some catchments when comparing ESP-SPI to the base ESP ensembles (mainly at longer lead times), but that, in general, we gain in sharpness. The loss in reliability does not necessarily mean that the ensemble becomes "unreliable". As illustrated in Figure 5, ESP-SPI ensembles are still not far from the diagonal of perfect reliability of the PIT diagram. Here again this relates to one of our objectives: how can we obtain sharper predictions while still having reliable ensembles.
* * *
RC: 3) The conditioning improved the performance of HistQ ensembles in forecasting low flow events and variables, but the conditioned ensembles were still less skilful than the Sys4 and ESP/ESP_SPI3 ensembles.

AR: The reviewer is right. If we look at Figure 6, HistQ_SPI3 appears to be less skilful than Sys4, ESP and ESP_SPI3 in terms of overall performance. Nevertheless, this ensemble has characteristics of interest for low-flow forecasting. In Figure 9, this ensemble systematically is in the best category for deficit duration, probably because it represents recessions better than the model does. This is quite an advantage of this ensemble. Again, we were interested in studying this ensemble HistQ because it can be a benchmark and a simple approach if a hydrological model is not used. We believe that investigating the possible ways of improving the HistQ approach is useful and, notably, provides insights to how the model (and its performance) influences (for better or worse) the quality of streamflow predictions.
* * *
RC: 4) The authors state that the ESP_SPI3 approach "systematically appears to be one of the best options to forecast deficit volumes." However, this conclusion is very subjective, as it is not authoritatively substantiated by the results presented in Figures 9 and 10.

AR: Figure 9 shows the results for the deficit duration, and not for the deficit in volumes. In Figure 10, which represents the reliability of the forecasting systems in terms of deficit volumes, ESP_SPI3 is the only forecasting system that belongs to the best Friedman category (non-parametric statistical test of significance) for the three lead times shown (category c at two weeks, and category b at five and twelve weeks) (these results are representative of the other lead times that are not shown). This is why we stated that it is the best method. We propose to replace "systematically" by "for all three lead times" to make the sentence clearer.
* * *
RC: Although several pages of this manuscript are spent discussing the results in great detail, and the authors walk through the discussion in a relatively clean, scientific manner, much of this discussion is centered around tangential topics. For example, the comparisons between the conditioned ESP/HistQ ensembles to the Sys4 ensembles seem irrelevant given that the conditioning did little to improve, and actually degraded in some cases, the skill compared to the base ensembles. Thus, comparing the conditioned ensembles to the Sys4 ensembles is equivalent to comparing the base ensembles to Sys4, which of course is unnecessary. The results should be restricted to and presented with the stated goal of the study in mind – improving the skill of historical observation-based ensemble forecasting systems.

AR: We will consider shortening the discussion to better highlight the key results of the paper.
* * *
RC: Unfortunately, because there is little to report on the utility of applying this conditioning method to seasonal streamflow and low flow forecasting, the authors may need to redesign and/or include other experiments before resubmitting this paper. One suggestion, actually offered by the authors, is to examine the utility of using SPEI to condition the ensembles. Although the SPI is likely sufficient to appropriately subset historical precipitation ensembles, it may not be sufficient from a streamflow perspective. It seems likely that the relative magnitude of an individual SPI value may not always be translated into a similar relative magnitude flow or volume value if ET is a major hydrologic control in the watershed of interest (i.e. late season streamflows can be very different following extended dry but mild vs extended dry but hot conditions). Thus, conditioning the ensembles with both precipitation- and temperature-driven indices may provide more robust results.

AR: We think that the systematic analysis that we have carried out and reported in the paper is of relevance for the community, mainly since several efforts have been recently put into improving meteorological seasonal forecasting systems to better quantify risks and impacts in hydrology. In this regard, we think that our paper provides useful insights to how hydrological seasonal forecasts can benefit from this information (when used directly as input to a hydrological model or when providing conditioning indices to select ensemble traces). It must be noted that the work carried out in this paper is based on previously bias corrected System 4 precipitation forecasts, as stated on line 7, Page 5 (see also our paper Crochemore et al., 2016). We thus investigate the performance of the conditioning considering an improved ensemble precipitation prediction system. We illustrate how different approaches have different limitations, but also different assets. In our opinion, this is an important contribution, notably to better meet operational expectations. We also think that our investigation on the utility of applying a conditioning method is useful to the community. We have evaluated the conditioning based on several major qualities of ensemble forecasts: overall performance, sharpness and reliability. We have demonstrated how these respond differently to the conditioning approaches. Again, it seems important to us that a developer or a user of seasonal forecasting systems be aware of the different impacts on forecast quality. We illustrate the performance of our ensembles with a low-flow forecasting case. We think that this illustration can also be useful to other users, with other preferences or operational focusses. Mainly, our study shows that the analysis of the usefulness of a forecasting system should not be restricted to evaluating some scores of forecast quality. It should also be extended to show how better forecasts impact the forecasting of the main variables of interest for a specific user and its decision-making context.

We agree with the reviewer that many other additional experiments could be done. For instance, the analysis of conditioning approaches based also on temperatures would be very interesting, although the impact of the potential evapotranspiration (used as input to the hydrological model) is expected to be of a second-order (as measures on the streamflow forecasts), comparatively to the impact of the rainfall conditioning, given that rainfall-runoff models are more sensitive to errors in rainfall than in potential evapotranspiration. The use of other indices could also be interesting for further studies. We could not add more experiments to our paper or it would become too long and lose its focus. We mentioned however numerous perspectives for further studies on Page 16 (line 10 onwards).
* * *
**RC:** Lastly, the underlying standard of this manuscript is the stated inherent reliability of historical observation-based ensembles, but this is a bit misleading. In true forecasting (not hindcasting), climatology-driven predictions may not be all that reliable. Several decades worth of historical information is often sought to build an ensemble forecasting system, but the climatic regime of the forecast area may be changing too rapidly for this. Thus, the distribution functions of actual forecasts and their corresponding observations may be offset from one another (i.e. not fall on a 1:1 line). Perhaps the authors should frame the goal more along the lines of using the conditioning to sharpen the ensembles, and less along the lines of marrying the reliability of historical ensembles with the sharpness of GCMs.

AR: We thank the reviewer for this interesting comment. We focused on the search for sharper ensembles while maintaining reliability, since this is a widespread notion in forecast verification. However, we also agree that the main message to convey is on using the conditioning to sharpen the ensembles (without deteriorating reliability). We will have this comment in mind when producing the revised version of the paper.
* * *
References:

Crochemore, L., M.-H. Ramos, F. Pappenberger, 2016: Bias correcting precipitation forecasts to improve the skill of seasonal streamflow forecasts. *Hydrol. Earth Syst. Sci.*, 20: 3601-3618

---

## Author Comment (AC2) · 19 Sep 2016

**Response to Reviewer#2**

The authors want to thank Reviewer#2 for the valuable comments, which will help us to enhance our paper. We provide below our answers to the comments.

**Reviewer 2**

This study proposes an approach to improve short- and long-range (10-90 days) streamflow forecasts by conditioning resampled historical observations based on ECMWF System 4 forecasts. The conditioning is applied on both precipitation and streamflow records. Results are compared with historical resampled streamflow and ensemble streamflow prediction (ESP) as reference forecasts. Overall, the paper is well written and provides good assessments of different model performances. Nevertheless, I am concerned with the proposed method to improve streamflow forecasts (selection of resampled data based on GCM forecasts) as well as the results (week performance of the proposed method). Therefore, I think the paper is not ready for publication and requires major revision.

Authors' reply (AR): We thank the reviewer for the evaluation. Our aim was also to demonstrate through an extensive analysis the limitations and assets of the different conditioning approaches, notably when looking at the main attributes of forecast quality that are often searched by developers and users of forecasting systems (i.e., overall performance as measured by the CRPS, reliability and sharpness). We think that our paper provides useful insights to how hydrological seasonal forecasts can benefit from conditioning information. Our study also shows that the analysis of the usefulness of a forecasting system should not be restricted to evaluating some scores of forecast quality. It should also be extended to show how better forecasts impact the forecasting of the main variables of interest for a specific user and its decision-making context (in our paper, low-flow forecasting). In this regard, we think that, even if weak performance of seasonal forecasts is often observed in mid-latitudes (as is the case of our study catchments in France), progress can be obtained by reporting on experiments that focus on trying to understand where benefits can be expected. We think that the reviewers' comments received on this paper will greatly help us to improve our paper for its potential future publication.
* * *
**Reviewer's comment (RC):** Major comments:
1) The manuscript states that (P4, L9) the aim of this study is "to generate forecasts that benefit from the reliability of climatology-based ensembles and the sharpness of System 4 precipitation forecasts." First the proposed method does not seem to benefit from the sharpness of System 4, rather the reason for increased precision (sharpness) in the conditioned forecasts is due to the reduced ensemble size which is independent of the System 4's degree of uncertainty. Second, the results (e.g. Figures 4-5) show that except for some marginal improvements in forecasts for short lead times (Figure 4 upper row), the proposed method degrade the performance of the reference methods (CRPSS and PITSS are negative). In several instances in the manuscript (such as P9, L17) the authors discuss the improvements to the sharpness of the forecasts using their conditioning approach while reliability and performance have declined compared to the reference methods which undermines the sharpness improvements. The authors state that "...the PIT diagrams at 45 days show that this decrease does not affect the overall reliability of the conditioned ensembles" This again shows that the proposed method has not been able to improve upon the conventional approaches.

Authors' reply (AR): Following also the comments of Reviewer #1 (see also our replies to Reviewer #1), we understand that the aims of our study need to be clarified. Our general aim is stated on lines 6-7, Page 4: "(…) to investigate how selecting historical data based on forecast precipitation indices contributes to the skill of seasonal streamflow forecasts". The aim stated on line 9-10, Page 4 ("The aim is to generate forecasts that benefit from the reliability of climatology-based ensembles and the sharpness of System 4 precipitation forecasts") refers to the aim behind selecting the conditioning approaches to investigate how these can improve seasonal hydrological prediction. We agree that this is not clear as stated in the paper and we will clarify it in the revised version.

It is also interesting to note that one of the results illustrated in the paper is the discussions one can have around the importance of having forecasts of improved reliability and sharpness. When the reviewer states that "reliability and performance have declined compared to the reference methods which undermines the sharpness improvements", we believe that this a point of view and an interesting topic of discussion: Why degrading reliability undermines improvements in sharpness? Is it overall true or does it depend on the hydrological application? Can a user be so interested in improving sharpness that he accepts the cost of losing a bit of reliability? We do not mean that ensembles do not need to be reliable, on the contrary, we believe this is a quality that we should preserve when bringing improvements to a probabilistic or ensemble-based forecasting system. But sometimes a compromise between reliability and sharpness needs to be reached, and this is part of the results we show here (see also our other paper Crochemore et al., 2016, recently published). We illustrate how different approaches have different limitations, but also different assets. In our opinion, this is an important contribution, notably to better meet operational expectations.
* * *
**RC:** 2) The proposed method selects forecast ensemble members based on their closeness to some statistics (P8, L17). The procedure to choose the number of ensemble members to keep, however, is not explained. Is the number of selected runs subjectively chosen? If so a sensitivity analysis needs to be conducted.

AR: For a given forecast period, the conditioning statistic is calculated for each member of the System 4 forecast. We thus have an ensemble of forecast statistics of the same size as the System 4 ensemble for the forecast period. For each member of this ensemble of forecast statistics, the closest historical scenario is identified and used as ensemble member (i.e. as a local temporal realization for that forecast statistic). We will clarify this in Section 2.4.2.
* * *
**RC:** 3) The method conditions the resampled precipitation and streamflow data to GCM forecasts. However, GCM forecasts are uncertain particularly at seasonal scales. That might explain why the overall results do not show improvements compared with conventional ESP. In particular, authors need to discuss how the method will perform in regions with high topographical variations (considering that the low-resolution GCMs cannot capture the regional hydroclimatic variations). Related to this please discuss why you compare the proposed conditioning approach (based on SYS4) with results of SYS4?

AR: The idea behind this conditioning is that, even though GCM forecasts are uncertain at seasonal scales, coarse precipitation statistics (such as the SPI or monthly sums) may be easier to predict than precipitation time series. The performance of System 4 in predicting these coarse statistics is presented in Figures 2 and 3. Based on these results, we could expect the conditioning to improve sharpness.
The idea behind the comparison with Sys4 was to evaluate how the conditioned ensembles resemble the forecasts directly derived from System 4 time series in terms of reliability and sharpness. Another idea was to check the added value of conditioning compared to using Sys4 alone. We propose to add a sentence at the beginning of Section 3.2.2 or 3.2.3 to clarify why we make the comparison.
* * *
**RC:** 4) Please clarify which are the statistics (section 2.4.2) calculated for each ECMWF ensemble member separately or for the average of the 51 ensemble runs?

AR: The statistics were calculated for each member so as to obtain an ensemble of statistics (see also our reply above). We will clarify this in the revised version.
* * *
**RC:** 5) P8, L25: "when directly selecting scenarios from past streamflow observations, the last observed streamflow is added as a conditioning criterion in the computation of the Euclidian distance." This is problematic as the last observed (previous year's(?)) streamflow is not a good indicator of the next year's streamflow in particular with regard to high and low flows which are driven by several hydroclimatic factors that do not necessarily repeat at consecutive years.

AR: In fact, the hydrological model is run at the daily time step and "the last observed streamflow" refers to the observed streamflow on the day of issuing the forecast (Section 2.2). We will make sure that this is clear in the revised version.

RC: 6) Resampled precipitation is considered to drive the hydrologic model, however, the mean interannual potential evapotranspiration is used instead of the resampled one. Considering that PET might have a substantial role in low flow forecasts, I recommend using the resampled PET as well.

AR: We used the mean multi-annual PET instead of the resampled one when conditioning ESP in order to compare it with System 4 streamflow forecasts. Indeed, System 4 streamflow forecasts are also produced by forcing the model with the mean multi-annual potential evapotranspiration.
As a matter of fact, we had first produced the results in Figures 5, 6 and 7 for the resampled PET (PET for the years resampled based on precipitation). The results we obtained were very close to those presented here.

RC: 7) P12, L12: "The rankings are based on the visual evaluation of Figure 5." Visual evaluation is not an appropriate ranking approach.

AR: For a more quantitative ranking, we will consider ranking the methods by using averaged skill scores in the revised version.

RC: 8) Results of section 3.4 are based on only one drought event for one catchment and cannot provide sufficient evidence for the overall performance of the methods.

AR: We agree; the aim of Section 3.4 is purely illustrative. We do not aim at providing a statistical assessment of overall performance, especially as the illustration refers to rare events in hydrologic risk assessment. We will clarify this and pay attention not to drawn any general conclusions on the statistical performance of the systems from the analysis of the figure.

RC: 9) P6, section 2.3.1 Please elaborate further on the differences between CRPS and PIT and how they should be interpreted when they show inconsistent results (e.g. Fig 4).

AR: The CRPS is the sum of several terms, one representing reliability and one being influenced by sharpness (Hersbach, 2000). Therefore, the CRPS can be stable even though reliability is deteriorated, provided that sharpness, for instance, is improved. We will add a few words in Section 2.3.1 in the revised version to clarify this.

RC: 10) Multi-model averaging methods (such as simple mean, Bayesian Model Averaging (BMA) etc.) (Duan et al. 2005, Najafi et al. 2015, Raftery et al. 2005) have shown to improve short and long term hydrologic forecasts. I would suggest discussing the application of these approaches to merge the ensemble of forecasts obtained from different methods in this study.

AR: This can be an interesting topic for further studies. We will consider it in the discussion/perspectives presented at the end of the paper in the revised version.

RC: Specific comments:

- Abstract "…forecasts based on GCM outputs can offer sharper ensembles… :": does "sharper" refer to more precise? Related to this please define "sharpness" and "reliability" before using these terms, in the Introduction.

AR: Sharper refers to the range of possible future scenarios. It is a property of the ensembles and do not depend on the observations (as is the case of accuracy). We will add short definitions to the concepts of sharpness and reliability in the revised version.
* * *
**RC:** - L15: ECMWF System 4: Please expand the full name.

AR: We will expand the full name in the revised version.
* * *
**RC:** - Abstract: "The four conditioned precipitation scenarios were used as input to the GR6J hydrological model to obtain eight conditioned streamflow forecast scenarios": The statement is vague as to how four precipitation scenarios result in eight streamflow scenarios?

AR: Indeed, we will rewrite this sentence to clarify the methodology.
* * *
**RC:** - P2, L19: ESP is one of the streamflow forecast methods which need to be discussed here. Also please note that in ESP all historical meteorological forcings can be resampled to run the hydrological model (not just precipitation as stated in LP2, L27)

AR: We see ESP as a hybrid approach: it is statistical in terms of precipitations (climatology) and dynamical when it comes to streamflow (referring to the use of a hydrological model). Therefore, we propose to add a sentence after "More importantly, some studies have shown that the two approaches can complement and benefit from each …" to better introduce the ESP as a type of combination of the two approaches. We will also change the definition of ESP to refer to meteorological forcing to a hydrological model.
* * *
**RC:** - P4, L3 Statement is not clear "although the ensemble conditioned from historical streamflows, which was the…"

AR: We propose to change this to "They found that the GCM-conditioned ensemble outperformed the ESP method. Nevertheless, the ensemble conditioned from historical streamflows was the most reliable. In addition, decisions based on that ensemble completely eliminated flood damage and generated more energy than decisions based on the other two ensembles."
* * *
**RC:** - P4, L12-15: Please move to the results section.

AR: We agree and will consider moving the text in the revised version.
* * *
**RC:** - P4, L17: Please define "discrimination"

AR: The discrimination of a system is its capacity to detect an event defined by a threshold. We will add a definition in Section 2.3.1, when presenting the ROC score.
* * *
**RC:** - P5, L3: Please explain how many grid cells lie within each catchment in average. How was the aggregation performed? Please also indicate the forecast starting date.

AR: Each catchment is covered by one to four grid cells. The aggregation method is a simple weighted mean of precipitations from different grid cells, based on the area of the catchment covered by each cell. Forecasts are issued for the 1$^{st}$ of each month. We will clarify this in the revised version.

**RC:** - P5, L23: What do you mean by "systematically"?

AR: We meant that ESP_SPI3 is the only forecasting system that belongs to the best Friedman category for the three lead times (category c at two weeks, and category b at five and twelve weeks). We will remove "systematically" and state explicitly that we refer to "all studied lead times".
* * *
**RC:** - P5, L31-33: What is the range of KGE values? Please show the equations for KGE and 1-bias and include their ranges.

AR: We will add the range of KGE values. We will also explain the way the bias was computed. However, we would prefer to avoid adding the equations for these two criteria since they are only mentioned once and a reference article is already provided for the KGE.
* * *
**RC:** - P6, L9: Please change "The CRPS averages over the evaluation period the area between the cumulative forecast distribution…" to "The CRPS averages the area between the cumulative forecast distribution… over the evaluation period." Similarly, for L12.

AR: We will correct this.
* * *
**RC:** - P7, L3: What is the "reference"? Is it HisQ? Please define.

AR: We will add the information.
* * *
**RC:** - I suggest bringing section 2.4 before section 2.3.

AR: We will consider changing the position of these two sections in the revised version.
* * *
**RC:** - Figure 2: What is the difference between SUM1-3 and SUM3

AR: SUM3 is the sum of precipitations over the 3-month forecast horizon. SUM1-1 corresponds to the sum of precipitations over the first month of the forecast horizon, SUM1-2 the second month and so on. We will clarify this in the revised version.
* * *
**RC:** - P9, L1 "The reference forecast used to compute the skill scores is historical precipitations (i.e. climatology)": Do you mean hydrologic model simulation driven by historical precipitation?

AR: The reference here is historical precipitations. The analysis refers to precipitations only and not to hydrological model simulations. We evaluate precipitation indices derived from GCM-outputs and compare them to the precipitation indices derived from all historical years of precipitation. In other words, we compare the performance of the precipitation inputs used to obtain System 4 streamflow forecasts, to the performance of the precipitation inputs used to obtain ESP.
* * *
**RC:** - P9, L3 "SPI forecasts issued from System 4 are reliable overall and in standard precipitation conditions" please provide a reference

AR: This sentence is based on the analysis of Figure 3, which we will explicitly cite in the revised version.
* * *
**References**

Crochemore, L., M.-H. Ramos, F. Pappenberger, 2016: Bias correcting precipitation forecasts to improve the skill of seasonal streamflow forecasts. *Hydrol. Earth Syst. Sci.*, 20: 3601-3618

Hersbach, Hans. "Decomposition of the Continuous Ranked Probability Score for Ensemble Prediction Systems." *Weather and Forecasting* 15, no. 5 (2000): 559–70.

---

## Author Response (AR1)

**Response to the Editor and the Reviewers**

The authors want to thank the Editor and Reviewers #1 and #2 for their valuable comments. Their different insights helped us to enhance the paper, better clarify our objectives and highlight the contribution of our results to the literature.

The main improvement brought to the revised version of our paper concern a needed clarification of the aim of the paper. In the revised version, we are better highlighting the main contribution of the paper towards the investigation of the impacts of conditioning strategies on different forecast attributes. We made it clearer that we are not looking after a conditioning strategy that is the best solution for the studied catchments in hydrological seasonal forecasting. To clarify our aim, we did the following:

- We clarified our purposes in the Abstract and Introduction. Namely, we reviewed Section 1.2 of the Introduction to clarify that we do not search for the best conditioning method, but rather we aim for a better understanding of how forecast attributes (such as reliability, sharpness, discrimination) are affected by conditioning. We rearranged the literature review to better reflect that, and we made clearer the aim of the study in Section 1.3: Scope of the study.
- We changed the parts of the text where the word "comparison" was giving the wrong idea of "searching for the best method". In our paper, all comparisons of performance were made with the aim of understanding the impacts on forecast attributes. We think that the changes we made, mainly in the titles of the sessions, now help in putting the reader on the right focus of our study.
- We changed the conclusion section and some parts of the analysis of the results to better highlight the results that are relevant to the objective of the paper

We provide below our detailed answers to the comments received.

**Editor's comments (ED):**

Based on two thoughtful reviews and my own impressions, I suggest major revisions before resubmission for a subsequent review cycle. I agree with the reviewers that the results are quite equivocal about the relative merits of the tested approaches. This outcome is of course publishable, and possibly useful to the field, provided the authors draw conclusions that are consistent with the results. Thus, the possible conclusion that the methods adopted do not, in fact, robustly improve the baseline/reference forecasts, or even degrade them, must be given some consideration by the authors -- this seems true beyond the first month, at least.

Authors' reply (AR): We clarified our objectives. We do not search for a conditioning method that is better (according to all possible forecast attributes) than a reference or baseline. Instead, we propose to reflect on how methods such as conditioning-based methods can affect forecast attributes that one is searching to improve. We hope that this is clarified in the revised version and shed lights to the contribution of the results and conclusions presented in the paper to the literature.

ED: I also urge the authors to consider the suggestion regarding conditioning with temperature or other variables, as this is more skillfully predicted by the GCM, and may give more positive results. I realize that this would require significant work, however, and let the authors judge whether it would be possible for this paper.

AR: We fully agree that investigating a conditioning based on the SPEI would be valuable if one is searching to improve forecast performance. However, following the clarification of our objectives, we believe that it is clearer now that this is not a crucial issue to achieve the objectives of the paper. As well mentioned by the Editor, this would require substantial additional work and additional analyses, to a paper that is already rather long. We also believe that our main conclusions on the impact of conditioning on forecast attributes would still be valid, given the characteristics of the studied catchments and the hydrological model used. Therefore, we did not add SPEI in the revised version, but we have kept the sentence that mentions this interesting perspective for further studies in the conclusion section.

```
ED: In addition, I think that there are several curious features of the results
that warrant further explanation. In particular, in Figure 6, I'd like the authors
to give a more thorough analysis of the flipflop in skill within the first 15 days.
A second suggestion is that the use of the last obs as a conditioning factor for
the hist-based ensembles may not be a bad idea, but obscures the impact of the
conditioning on other factors, which is of interest (and more comparable to the
Sys4 based ensembles).
```

AR: In Fig. 6, we believe that there might be two effects affecting the evolution of performance and causing the flipflop at the first lead times. This may be partly influenced by initial conditions and its impact over time, and partly by how the catchments respond to precipitation. As the initial conditions are common to both systems (the systems assessed and the reference), the second influence might be playing a more prominent role. The quality of the precipitation forecasts from Syst4 at these short scales, together with the response of the catchments to the forcing, may be responsible for a low performance of the reference, causing the high values of CRPSS, for instance. We searched on the literature and found that a similar effect was reported by Brown, 2013 (see here: http://www.nws.noaa.gov/oh/hrl/hsmb/docs/hep/publications_presentations/Contract_2012-04-HEFS_Deliverable_02_Phase_I_report_FINAL.pdf) On page 39, they report on how "(…) the CRPSS increases during the first day, peaks with the residual contribution from the MEFP-GFS and declines thereafter.". This is the only reference we found that reports on this issue. We note that daily evaluations of seasonal forecasts are more difficult to find in the literature, as authors usually aggregate data at the weekly or monthly time steps. We think the flipflop is an interesting issue but it is out of the scope of our paper to investigate it in details.

Concerning the use of the last observation as a conditioning factor, we agree that it might favours the evaluation of the ensembles based on historical streamflows, but we think it would be too damaging to the analysis to impose hydrological conditions in the forecasting that could be obviously unrealistic given the daily time step we are using in the analysis (from a day to the other, river flows may be very different, regardless the fact that they are recorded in the same season). We believe however that the impact of this conditioning on our main conclusions is limited to the shorter lead times.

**Reviewer 1**

```
As an outsider to the professional academic world, I feel that I cannot speak with
unquestionable credibility to the novelty or scientific soundness of this
manuscript – I am simply not familiar enough with the wealth of recent research
into seasonal hydrologic forecasting. However, I can supply my overall impression
of this work, which may be useful given my background in operational hydrologic
forecasting.
```

```
Reviewer's comment (RC): The authors reference several studies that utilized
approaches similar to the one undertaken here – conditioning historical
observation-based ensembles to improve forecasts generated from these ensembles.
```

Thus, the fundamental direction of the current study is not overly original.
However, the manner in which the conditioning was applied – using GCM- and
climatology-derived precipitation indices to select the most relevant historical
ensembles – does appear to be a novel approach.

Authors' reply (AR): We thank the reviewer for the comment. We agree that studies that use GCM-derived precipitation indices as conditioning indices are rare and applications of conditioning approaches over mid-latitudes (in our case, France), where the reliability of seasonal weather predictions is, in general, low, are important contributions to the community. Additionally, in the revised version, we believe that the objectives of our paper are clarified. We highlight that the novel aspect of the paper is not on simply using conditioning approaches and trying to improve over a reference forecast, but it relies on scrutinizing the analysis of their effects on different forecast attributes. We want to draw attention to the fact that it is not straightforward to find a method that is best in all quality attributes of an ensemble prediction, whether we are looking for overall performance, sharpness, reliability or discrimination for capturing extreme events such as droughts.
* * *
**RC:** The potential utility of this approach is presented well in Figure 2, where the precipitation indices generated from the GCM hindcasts (ECMWF Sys4) are compared against those generated from the historical observations. As the authors state, the Sys4 indices perform at least as well as the base indices overall (CRPSS), even outperform at one month lead time, but are consistently sharper (IQRSS). Further, the Sys4 indices have good reliability overall (Figure 3). The reliability of the indices falters when looking at only drier than normal or only wetter than normal conditions, but this seems to be unavoidable with any forecasting approach.
Despite the prefaced potential of using the Sys4 precipitation indices to condition, or subset, historical ensembles, this study's results offer just marginal practical insight:

1) Subsetting the ensembles based on the precipitation indices improve the HistQ performance more than it does the ESP performance. This result is not very useful, however, since the HistQ approach is rudimentary (and likely rarely used), and the primary benefit of the conditioning is seen during short lead times (which is simply the effect of blending from the last streamflow observation).

AR: The reviewer has clearly understood the aims of the analysis illustrated in Figure 2 and we acknowledge the positive comment provided. We believe this first step in analysing the performance of the precipitation-based conditioning indices is essential previously to analysing the conditioned outputs in terms of streamflow, given the non-linearity in the transformation of precipitation into streamflow in a hydrological model.
Also, we included HistQ in our study because this is a "poorman's approach" that can serve as a naïve benchmark, where no hydrological model but only a long streamflow time series of records is available. This comment was added in Section 2.3.1 when presenting HistQ. One of the objectives of our study was to see whether this rudimentary approach could be turned into a valuable one provided that precipitation anomalies are available. In this sense, the improvement observed in sharpness, for instance, as it lasts over longer lead times, is mostly due to the conditioning. This improvement is one of the aims of several operational seasonal hydrological prediction systems: obtain sharper predictions, while maintaining reliability.
The revised version has clearer statements on our objectives, and so we believe that the practical insights of our analyses are now better highlighted.
* * *
**RC:** 2) For ESP, SPI-conditioning appears to outperform SUM-conditioning, but this statement is qualitative at best and neither set of conditioned ensembles provides any notable improvements over the base ensembles. Compared to the base ESP ensembles, the sharpness of the ESP_SPI3 ensembles was improved by up to 10% but the reliability was degraded by up to 40% (Figure 7).

AR: We thank the reviewer for these comments, which led us to review carefully the section presenting the results from SPI-conditioning and SUM-conditioning. The analysis is illustrated in Figure 5. We can see that conditioning on SPI (third and fourth columns) provides better scores over

(or at least do not degrade the score of) the reference (base ESP ensembles) than conditioning with SUM (first and second columns). For instance, based on the IQRSS results, we can see that the SUM-conditioning may decrease sharpness in some cases, whereas the SPI-conditioning guarantees to maintain or even increase sharpness. Based on the PIT diagram analysis, the SUM-conditioning causes an overprediction of observations, whereas the SPI-conditioning clearly limits this effect. Figure 7, mentioned by the reviewer, provides indeed the means for a more quantitative analysis. However, it must be noted that it is restricted to the results for SPI-conditioning. In Figure 7, we can see that we can lose on the score for reliability (PIT area) at some catchments when comparing ESP-SPI to the base ESP ensembles (mainly at longer lead times), but that, in general, we gain in sharpness. The lost in terms of score of reliability does not necessarily mean that the ensemble becomes "unreliable". As illustrated in Figure 5, ESP-SPI ensembles are still not far from the diagonal of perfect reliability of the PIT diagram. We think these analyses are useful to illustrate our main objective in this paper: better assess (and understand) how conditioning affects different attributes of forecast quality.

To clarify this issue, we highlighted the aims of the paper in the revised version, and we deleted some misleading sentences and assertions from the description of the results.
* * *
**RC:** `3) The conditioning improved the performance of HistQ ensembles in forecasting low flow events and variables, but the conditioned ensembles were still less skilful than the Sys4 and ESP/ESP_SPI3 ensembles.`

AR: The reviewer is right, and more can be added to the analyses. In fact, if we look at Figure 6, HistQ_SPI3 appears to be less skilful than Sys4, ESP and ESP_SPI3 in terms of overall performance. Nevertheless, this ensemble has characteristics of interest for low-flow forecasting. In Figure 9, this ensemble is systematically in the best category for deficit duration, probably because it represents recessions better than the hydrological model does. This is quite an advantage of this ensemble based on historical observations. Again, we were interested in studying this ensemble HistQ because it can be a benchmark and a simple approach if a hydrological model is not used. We believe that investigating the possible ways of improving the HistQ approach is useful and, notably, provides insights to how the model (and its performance) influences (for better or worse) the quality of streamflow predictions. We think that by clarifying our objectives and better focusing our results towards these useful insights, we made our point clearer in the revised version.
* * *
**RC:** `4) The authors state that the ESP_SPI3 approach "systematically appears to be one of the best options to forecast deficit volumes." However, this conclusion is very subjective, as it is not authoritatively substantiated by the results presented in Figures 9 and 10.`

AR: We had placed the statement based on Figure 10 only, which is the one that represents the reliability of the forecasting systems in terms of deficit in volume (Fig. 9 is for deficit in duration). In the revised version, we replaced "systematically" by "for all three lead times" to make the sentence clearer and avoid confusion.
* * *
**RC:** `Although several pages of this manuscript are spent discussing the results in great detail, and the authors walk through the discussion in a relatively clean, scientific manner, much of this discussion is centered around tangential topics. For example, the comparisons between the conditioned ESP/HistQ ensembles to the Sys4 ensembles seem irrelevant given that the conditioning did little to improve, and actually degraded in some cases, the skill compared to the base ensembles. Thus, comparing the conditioned ensembles to the Sys4 ensembles is equivalent to comparing the base ensembles to Sys4, which of course is unnecessary. The results should be restricted to and presented with the stated goal of the study in mind – improving the skill of historical observation-based ensemble forecasting systems.`

AR: We thank the review for this comment that helped us to better focus the presentation of the aims of the paper and of our results. We rewrote Sections 3.2.2 and 3.2.3 in the light of the new clarified objectives. The aim of the comparison with Sys4 is to see which forecast attributes of System 4 got "transferred" to the conditioned ensembles and which did not. Section 3.2.2 was substantially reduced

to better focus on this aspect: ESP and HistQ are no longer compared to Sys4, and Figure 6 was limited to the comparison between the conditioned ensembles and System 4. We hope that the new presentation of the results also clarifies our general aim.
* * *
**RC:** Unfortunately, because there is little to report on the utility of applying this conditioning method to seasonal streamflow and low flow forecasting, the authors may need to redesign and/or include other experiments before resubmitting this paper. One suggestion, actually offered by the authors, is to examine the utility of using SPEI to condition the ensembles. Although the SPI is likely sufficient to appropriately subset historical precipitation ensembles, it may not be sufficient from a streamflow perspective. It seems likely that the relative magnitude of an individual SPI value may not always be translated into a similar relative magnitude flow or volume value if ET is a major hydrologic control in the watershed of interest (i.e. late season streamflows can be very different following extended dry but mild vs extended dry but hot conditions). Thus, conditioning the ensembles with both precipitation- and temperature-driven indices may provide more robust results.

AR: The revised version of the paper clarifies our objectives, which are not towards finding the best conditioning method. It seems important to us that a developer or a user of a seasonal forecasting system be aware of the different impacts a forecasting method may have on forecast quality, regardless of the more or less sophisticated conditioning the user is using or developing. We believe that, by clarifying our focus, the revised paper shows our results in a different light and, in this sense, the addition of more conditioning methods based on SPEI, although interesting in an inter-comparison analysis, becomes less relevant for the investigation we propose on the impacts on forecast attributes. We recognize it as an interesting further aspect to be investigated and thus kept the sentence drawing attention to it in the conclusion section (see also our reply above to the Editor's comments).
* * *
**RC:** Lastly, the underlying standard of this manuscript is the stated inherent reliability of historical observation-based ensembles, but this is a bit misleading. In true forecasting (not hindcasting), climatology-driven predictions may not be all that reliable. Several decades worth of historical information is often sought to build an ensemble forecasting system, but the climatic regime of the forecast area may be changing too rapidly for this. Thus, the distribution functions of actual forecasts and their corresponding observations may be offset from one another (i.e. not fall on a 1:1 line). Perhaps the authors should frame the goal more along the lines of using the conditioning to sharpen the ensembles, and less along the lines of marrying the reliability of historical ensembles with the sharpness of GCMs.

AR: We thank the reviewer for this interesting comment. We focused on the search for sharper ensembles while maintaining reliability, since this is a widespread notion in forecast verification. However, we also agree that the main message to convey is on using the conditioning to sharpen the ensembles (without deteriorating reliability). By clarifying our objectives and better introducing our results, we believe that we now avoid this pitfall in the revised version.

**Reviewer 2**

This study proposes an approach to improve short- and long-range (10-90 days) streamflow forecasts by conditioning resampled historical observations based on ECMWF System 4 forecasts. The conditioning is applied on both precipitation and streamflow records. Results are compared with historical resampled streamflow and ensemble streamflow prediction (ESP) as reference forecasts. Overall, the paper is well written and provides good assessments of different model performances. Nevertheless, I am concerned with the proposed method to improve streamflow forecasts (selection of resampled data based on GCM forecasts) as well as the results (week performance of the proposed method). Therefore, I think the paper is not ready for publication and requires major revision.

Authors' reply (AR): We thank the reviewer for this evaluation. We have now clarified our aim in the revised version: we do not search to "propose a method to improve streamflow forecasts", but we aim at investigating how methods (and particularly, conditioning-based methods) affect the evaluation of forecast quality according to different attributes (highlighting existing inter-dependencies). For this we carried out the extensive analysis proposed in the paper, investigating the limitations and assets of the different conditioning approaches, notably when looking at the main attributes of forecast quality that are often searched by developers and users of forecasting systems (i.e., overall performance as measured by the CRPS, reliability and sharpness, and discrimination for low flow events). Our paper provides useful insights to how hydrological seasonal forecasts can benefit from conditioning information. Our study also shows that the analysis of the usefulness of a forecasting system should not be restricted to evaluating some scores of forecast quality. It should also be extended to show how better forecasts impact the forecasting of the main variables of interest for a specific user and its decision-making context (in our paper, low-flow forecasting). In this regard, we think that, even if only marginal improvements in the performance of the conditioned seasonal forecasts are observed, progress can be obtained by reporting on experiments that focus on trying to understand where benefits can be expected.

We think that our revised version now clarifies this point and better highlights the contribution of our paper to the literature.
* * *
**Reviewer's comment (RC):** Major comments:
1) The manuscript states that (P4, L9) the aim of this study is "to generate forecasts that benefit from the reliability of climatology-based ensembles and the sharpness of System 4 precipitation forecasts." First the proposed method does not seem to benefit from the sharpness of System 4, rather the reason for increased precision (sharpness) in the conditioned forecasts is due to the reduced ensemble size which is independent of the System 4's degree of uncertainty. Second, the results (e.g. Figures 4-5) show that except for some marginal improvements in forecasts for short lead times (Figure 4 upper row), the proposed method degrade the performance of the reference methods (CRPSS and PITSS are negative). In several instances in the manuscript (such as P9, L17) the authors discuss the improvements to the sharpness of the forecasts using their conditioning approach while reliability and performance have declined compared to the reference methods which undermines the sharpness improvements. The authors state that "...the PIT diagrams at 45 days show that this decrease does not affect the overall reliability of the conditioned ensembles" This again shows that the proposed method has not been able to improve upon the conventional approaches.

Authors' reply (AR): Our aim was not to propose a method that would improve over a reference or baseline. We agree that this was not clear in the original paper and we have clarified this issue in the revised version (see also our replies above to Reviewer #1 and the Editor). As illustrated by the reviewer's comment, the discussion on improving sharpness and reliability is an interesting one, which attracts attention. We believe that reliability is an attribute of forecast quality that we should preserve when bringing improvements to a probabilistic or ensemble-based forecasting system. However, we try to show in the paper that sometimes a compromise between improving reliability and sharpness needs to be reached, and this is part of the results we show here (see also our other paper Crochemore et al., 2016, recently published). We illustrate how different approaches have different limitations, but also different assets. In our opinion, this is an important contribution, notably to better meet operational and developers' expectations.
* * *
**RC:** 2) The proposed method selects forecast ensemble members based on their closeness to some statistics (P8, L17). The procedure to choose the number of ensemble members to keep, however, is not explained. Is the number of selected runs subjectively chosen? If so a sensitivity analysis needs to be conducted.

AR: For a given forecast period, the conditioning statistic is calculated for each member of the System 4 ensemble forecast. We thus have an ensemble of forecast statistics of the same size as the System 4 ensemble for the forecast period. For each member of this ensemble of forecast statistics, the closest historical scenario is identified and used as an ensemble member in the methods investigated. We clarified this in Section 2.3.2 of the revised version.

RC: 3) The method conditions the resampled precipitation and streamflow data to GCM forecasts. However, GCM forecasts are uncertain particularly at seasonal scales. That might explain why the overall results do not show improvements compared with conventional ESP. In particular, authors need to discuss how the method will perform in regions with high topographical variations (considering that the low-resolution GCMs cannot capture the regional hydroclimatic variations). Related to this please discuss why you compare the proposed conditioning approach (based on SYS4) with results of SYS4?

AR: The idea behind this conditioning is that, even though GCM forecasts are uncertain at seasonal scales, coarse precipitation statistics (such as the SPI or monthly sums) may be easier to predict than precipitation time series. The performance of System 4 in predicting these coarse statistics is presented in Figures 2 and 3. Based on these results, we could expect the conditioning to improve sharpness.
The idea behind the comparison with Sys4 was to evaluate how the conditioned ensembles resemble the forecasts directly derived from System 4 time series in terms of reliability and sharpness. Another idea was to check the added value of conditioning compared to using Sys4 alone. In the revised version, we rewrote Sections 3.2.2 and 3.2.3 to clarify our objectives (which are not to propose a better method, but to investigate impacts on attributes of forecast quality; see our replies above) and we hope we have clarified these issues.

RC: 4) Please clarify which are the statistics (section 2.4.2) calculated for each ECMWF ensemble member separately or for the average of the 51 ensemble runs?

AR: The statistics were calculated for each member so as to obtain an ensemble of statistics (see also our reply above). We clarified this in the revised version (Section 2.3.2).

RC: 5) P8, L25: "when directly selecting scenarios from past streamflow observations, the last observed streamflow is added as a conditioning criterion in the computation of the Euclidian distance." This is problematic as the last observed (previous year's(?)) streamflow is not a good indicator of the next year's streamflow in particular with regard to high and low flows which are driven by several hydroclimatic factors that do not necessarily repeat at consecutive years.

AR: In fact, the hydrological model is run at the daily time step and "the last observed streamflow" refers to the observed streamflow on the day of issuing the forecast (Section 2.2). This was clarified in the revised version (Section 2.3.2).

RC: 6) Resampled precipitation is considered to drive the hydrologic model, however, the mean interannual potential evapotranspiration is used instead of the resampled one. Considering that PET might have a substantial role in low flow forecasts, I recommend using the resampled PET as well.

AR: We used the mean multi-annual PET instead of the resampled one when conditioning ESP in order to compare it with System 4 streamflow forecasts. Indeed, System 4 streamflow forecasts are also produced by forcing the model with the mean multi-annual potential evapotranspiration. We have checked the results in Figures 5, 6 and 7 for the resampled PET (PET for the years resampled based on precipitation), and the results we obtained were very close to those presented in the paper.

**RC:** 7) P12, L12: "The rankings are based on the visual evaluation of Figure 5." Visual evaluation is not an appropriate ranking approach.

AR: The reviewer is right. For a more quantitative analysis, we ranked the methods based on the averaged skill scores in the revised version.
* * *
**RC:** 8) Results of section 3.4 are based on only one drought event for one catchment and cannot provide sufficient evidence for the overall performance of the methods.

AR: We fully agree with the reviewer. The aim of Section 3.4 is purely illustrative and we clarified this in the revised version. We notably paid attention not to draw any general conclusions on the statistical performance of the systems from the analysis of the figure.
* * *
**RC:** 9) P6, section 2.3.1 Please elaborate further on the differences between CRPS and PIT and how they should be interpreted when they show inconsistent results (e.g. Fig 4).

AR: The CRPS is the sum of several terms, one representing reliability and one being influenced by sharpness (Hersbach, 2000). Therefore, the CRPS can be stable even though reliability is deteriorated, provided that sharpness, for instance, is improved. In the revised version, we added some sentences in Section 2.4.1 to clarify this.
* * *
**RC:** 10) Multi-model averaging methods (such as simple mean, Bayesian Model Averaging (BMA) etc.) (Duan et al. 2005, Najafi et al. 2015, Raftery et al. 2005) have shown to improve short and long term hydrologic forecasts. I would suggest discussing the application of these approaches to merge the ensemble of forecasts obtained from different methods in this study.

AR: This can be an interesting topic for further studies. We added a sentence on this perspective in the conclusion section of the revised version.
* * *
**RC:** Specific comments:

- Abstract "…forecasts based on GCM outputs can offer sharper ensembles… :": does "sharper" refer to more precise? Related to this please define "sharpness" and "reliability" before using these terms, in the Introduction.

AR: Sharper refers to the range of possible future scenarios. It is a property of the ensembles and do not depend on the observations (as is the case of accuracy). We added short definitions to the concepts of sharpness and reliability in Section 1.2 of the Introduction in the revised version.
* * *
**RC:** - L15: ECMWF System 4: Please expand the full name.
    - Abstract: "The four conditioned precipitation scenarios were used as input to the GR6J hydrological model to obtain eight conditioned streamflow forecast scenarios": The statement is vague as to how four precipitation scenarios result in eight streamflow scenarios?

AR: We corrected these issues in the revised version.
* * *
**RC:** - P2, L19: ESP is one of the streamflow forecast methods which need to be discussed here. Also please note that in ESP all historical meteorological forcings can be resampled to run the hydrological model (not just precipitation as stated in LP2, L27)

AR: Following the reviewer's comment, ESP is now discussed in this section in the revised version. We also paid attention to refer to all the meteorological forcings to a hydrological model rather than just precipitation.
* * *
RC: - P4, L3 Statement is not clear "although the ensemble conditioned from historical streamflows, which was the…"
    - P4, L12-15: Please move to the results section.

AR: This section was rewritten in the revised version and these sentences were modified and moved in the process, following the reviewer's comments.
* * *
RC: - P4, L17: Please define "discrimination"

AR: The discrimination of a system is its capacity to detect an event defined by a threshold. We added this definition in Section 2.4.1, when presenting the ROC score.
* * *
RC: - P5, L3: Please explain how many grid cells lie within each catchment in average. How was the aggregation performed? Please also indicate the forecast starting date.

AR: Each catchment is covered by two to ten grid cells. The aggregation method is a simple weighted mean of precipitations from different grid cells, based on the area of the catchment covered by each cell. Forecasts are issued for the 1$^{st}$ of each month. We clarified this in the revised version (Section 2.1).
* * *
RC: - P5, L23: What do you mean by "systematically"?

AR: We meant that, regardless of the forecast year, the mean potential evapotranspiration is used as input to the hydrological model. We replaced "systematically" by "regardless of the forecast year" to be more precise and avoid confusion (Section 2.2).
* * *
RC: - P5, L31-33: What is the range of KGE values? Please show the equations for KGE and 1-bias and include their ranges.

AR: We added the ranges of KGE values to Section 2.2. We also added a comment on the way the bias was computed. However, we would prefer to avoid adding the equations for these two criteria since they are only mentioned once and a reference article is already provided for the KGE.
* * *
RC: - P6, L9: Please change "The CRPS averages over the evaluation period the area between the cumulative forecast distribution…" to "The CRPS averages the area between the cumulative forecast distribution… over the evaluation period." Similarly, for L12.

AR: We corrected this in the revised version (Section 2.4.1).
* * *
RC: - P7, L3: What is the "reference"? Is it HisQ? Please define.

AR: We clarified this point in the text and now explicitly cite the references used in the article as base ensembles (Section 2.4.2).
* * *
RC: - I suggest bringing section 2.4 before section 2.3.

AR: Following the reviewer's recommendation, we moved Section 2.4 before Section 2.3 so that ensemble forecasts are presented before the methods used to evaluate them.
* * *
RC: - Figure 2: What is the difference between SUM1-3 and SUM3

AR: SUM3 is the sum of precipitations over the 3-month forecast horizon. SUM1-1 corresponds to the sum of precipitations over the first month of the forecast horizon, SUM1-2 the second month and so on. We detailed this in the legend of Figure 2. We also added a short note in Section 2.3.2 when describing the conditioned scenarios.
* * *
RC: - P9, L1 "The reference forecast used to compute the skill scores is historical precipitations (i.e. climatology)": Do you mean hydrologic model simulation driven by historical precipitation?

AR: The reference here is historical precipitations. The analysis refers to precipitations only and not to hydrological model simulations. We evaluate precipitation indices derived from GCM-outputs and compare them to the precipitation indices derived from all historical years of precipitation. In other words, we compare the performance of the precipitation inputs used to obtain System 4 streamflow forecasts, to the performance of the precipitation inputs used to obtain ESP.
* * *
RC: - P9, L3 "SPI forecasts issued from System 4 are reliable overall and in standard precipitation conditions" please provide a reference

AR: This sentence is based on the analysis of Figure 3. We explicitly cited Figures 2 and 3 in the appropriate sentences in the revised version (Section 3.1).
* * *
**References**

[revised manuscript text omitted]

---

## Author Response (AR2)

**Response to the Editor**

The authors want to thank the Editor for his comments on the revised version of our manuscript and the clarifications he suggested. The technical modifications in the new version are:

1) We modified "reversely" to "conversely" in the abstract, as well as in section 3.2.3

2) The information in the last sentence was mainly in "over a wider range of lead times", but we did modify the sense to clarify its point. The end of the abstract now reads:

> "The impact of conditioning was assessed in terms of forecast sharpness (spread), reliability, overall performance and low-flow event detection. Results showed that conditioning past observations on seasonal precipitation indices generally improves forecast sharpness, but may reduce reliability, with respect to climatology. Conversely, conditioned ensembles were more reliable but less sharp than streamflow forecasts derived from System 4 precipitations. Forecast attributes from conditioned and unconditioned ensembles are illustrated for a case of drought risk forecasting: the 2003 drought in France. In the case of low-flow forecasting, conditioning results in ensembles that can better assess weekly deficit volumes and durations over a wider range of lead times."

3) We added one general conclusion stating that

> "-        The use of Sys4 forecasts to derive conditioned ensembles generally did not improve the overall performance of seasonal streamflow forecasts. Overall performance criteria typically give equal weight to complementary features of forecast quality. This is the case of the CRPS, which, unless specified otherwise, gives equal weights to its components of reliability, resolution and uncertainty (Pappenberger et al., 2015)."

4) We added a short data availability section which we know is required to publish in HESS.

[revised manuscript text omitted]